# WHEN GNNS MEET SYMMETRY IN ILPS: AN ORBIT-BASED FEATURE AUGMENTATION APPROACH

Qian Chen[1,2], Lei Li[1,2], Qian Li[2], Jianghua Wu[2], Akang Wang[2,3,*], Ruoyu Sun[2,3], Xiaodong Luo[2], Tsung-Hui Chang[2,4], and Qingjiang Shi[2,5]

[1]School of Science and Engineering, The Chinese University of Hong Kong, Shenzhen, China
[2]Shenzhen International Center for Industrial and Applied Mathematics, Shenzhen Research Institute of Big Data, China
[3]School of Data Science, The Chinese University of Hong Kong, Shenzhen, China
[4]School of Artificial Intelligence, The Chinese University of Hong Kong, Shenzhen, China
[5]School of Software Engineering, Tongji University, Shanghai, China

## ABSTRACT

A common characteristic in integer linear programs (ILPs) is symmetry, allowing variables to be permuted without altering the underlying problem structure. Recently, GNNs have emerged as a promising approach for solving ILPs. However, a significant challenge arises when applying GNNs to ILPs with symmetry: classic GNN architectures struggle to differentiate between symmetric variables, which limits their predictive accuracy. In this work, we investigate the properties of permutation equivariance and invariance in GNNs, particularly in relation to the inherent symmetry of ILP formulations. We reveal that the interaction between these two factors contributes to the difficulty of distinguishing between symmetric variables. To address this challenge, we explore the potential of feature augmentation and propose several guiding principles for constructing augmented features. Building on these principles, we develop an orbit-based augmentation scheme that first groups symmetric variables and then samples augmented features for each group from a discrete uniform distribution. Empirical results demonstrate that our proposed approach significantly enhances both training efficiency and predictive performance.

## 1 INTRODUCTION

**Integer Linear Programs (ILPs)** are fundamental optimization problems characterized by a linear objective function and linear constraints, where the decision variables are restricted to integer values. These problems play a critical role in various fields, including operations research, computer science, and engineering (Pochet & Wolsey, 2006; Liu & Fan, 2018; Watson & Woodruff, 2011; Luathep et al., 2011; Schöbel, 2001). In practice, many ILPs exhibit a structural property known as **symmetry**, where certain permutations of decision variables leave both the problem structure and the solution set unchanged (Margot, 2003). For example, in the widely used collection of real-world ILPs, MIPLIB (Gleixner et al., 2021), approximately 32% of its instances display certain symmetries.

**Classic methods** for solving ILPs, such as branch-and-bound and branch-and-cut (Boyd & Mattingley, 2007; Morrison et al., 2016), systematically explore the solution space by breaking it down into smaller sub-problems and eliminating regions that do not contain optimal solutions. However, the presence of symmetry in these problems can result in redundant exploration of equivalent solutions, which hampers efficiency. To address this, several approaches have been developed. Margot (2002) prunes the enumeration tree in the branch and bound algorithm; Ostrowski et al. (2008; 2011) propose branching strategies based on orbits; Puget (2003; 2006) enhance problem formulations by introducing symmetry-breaking constraints. A more comprehensive survey of related research is

---

*Corresponding author: Akang Wang <wangakang@sribd.cn>

provided in (Margot, 2009). These techniques reduce the search space by detecting and removing symmetric solutions, allowing the solver to focus on unique, non-redundant parts of the problem. By leveraging symmetry in this manner, the overall efficiency and convergence of ILP solvers are significantly improved. While classic methods have been widely used, they fall short in terms of efficiency for real-world applications, calling for more advanced approaches.

**Recent advancements in machine learning (ML)** have opened new avenues for solving ILPs, offering approaches that enhance both efficiency and scalability (Gasse et al., 2022). Among these techniques, Graph Neural Networks (GNNs) have shown significant superiority in capturing the underlying structure of ILPs. By representing the problem as a graph, GNNs are able to exploit relational information between variables and constraints, which allows for more effective problem-solving strategies. Several categories of works have demonstrated the potential of GNNs in this context. Gasse et al. (2019) first proposed a bipartite representation of ILPs and applied GNNs to learn efficient branching decisions in branch-and-bound algorithms. Such graph representation was then utilized or enhanced by many subsequent researchers. Nair et al. (2020) utilized GNNs to predict initial assignments for ILP solvers to identify high-quality solutions. Khalil et al. (2022) integrated GNNs into the node selection process of the branch and bound framework. Other notable examples include learning to select cuts (Paulus et al., 2022), learning to configure (Iommazzo et al., 2020) and so on. A more comprehensive review of relevant works can be found in Cappart et al. (2023).

**Challenges:** Despite the growing number of ML-based methods for solving ILPs, only a few works have noticed the intrinsic property of ILPs—symmetry. This oversight often results in poor performance on problems with significant symmetries. Typical approaches, such as learning a GNN to predict the optimal solution, face the following *indistinguishability issue* when encountering symmetries in ILPs (a specific example is depicted in Figure 1):

*Traditional GNNs are incapable of distinguishing symmetric variables, limiting their effectiveness on ILPs with symmetry.*

**Symmetry-breaking in ML**: The issue of indistinguishability stems from the limitations of permutation-equivariant functions in handling data with inherent symmetries. Recent studies, particularly in chemistry and physics, have explored solutions to similar issues. One approach is to introduce augmented features into graph- or set-structured data to break symmetry. For example, Xie & Smidt (2024) introduces equivariant symmetry-breaking sets (SBS), which use symmetry groups to provide more informative inputs, breaking symmetry and improving computational efficiency. Similarly, Lawrence et al. (2024) extends SBS by incorporating probabilistic methods and canonicalization techniques for further efficiency. Morris et al. (2024) takes a different approach, using orbits for symmetry-breaking, offering a simple yet effective solution for graph data. Earlier works like Smidt et al. (2021), Zhang et al. (2021), and Kaba & Ravanbakhsh (2023) have laid the foundation for understanding symmetry in neural networks. In the ILP context, a few studies have also addressed this issue. Chen et al. (2022) tackles the problem of GNNs failing to distinguish foldable instances by adding random features to enhance expressiveness. Likewise, Han et al. (2023) and Chen et al. (2024) add positional embeddings to bipartite ILP representations, helping mitigate symmetry-related challenges.

**Motivation**: While the broader literature has extensively explored symmetry-breaking in chemical and physical systems, research on addressing indistingushability issue in ILPs remains limited. To date, no studies have fully adapted the advanced symmetry-breaking techniques used in these fields to ILP problems, nor have they leveraged the unique structural properties of ILPs to tackle symmetry more effectively. Moreover, there is a notable lack of theoretical analysis and empirical validation regarding the efficacy of existing machine learning methods for handling symmetry in ILPs. This gap highlights the urgent need for more robust, symmetry-aware solutions that can exploit the inherent symmetries of ILPs, ultimately improving the performance of ILP solvers on symmetric instances and making them more efficient in real-world applications.

**Contributions:** Considering the limitation of traditional GNNs in predicting the solutions for ILPs with symmetric variables, we first explore the inherent formulation symmetry property of these ILPs. By investigating the interplay between it and the permutation equivariance and invariance of GNNs, we show that they together lead to the performance limitation. To address it, we exploit feature augmentation and propose three guiding principles in constructing augmented features, including

i) distinguishability, ii) augmentation parsimony, and iii) isomorphic consistency. The first principle enables GNNs to output different values for symmetric variables and the second one avoids introducing 'conflict' training samples that could mislead GNNs to yield wrong predictions. Meanwhile, the second principle aims to keep the augmented features as simple as possible to enhance the training efficiency. Further, we devise an orbit-based feature augmentation scheme and analyze the difference between our proposed design and other existing schemes under these principles. Finally, our proposed orbit-based scheme is tested over classic ILP problems with significant symmetry and compared with existing schemes to validate the effectiveness of our proposed principles and design. Our contributions are summarized as follows.

- Theoretically demonstrating that the interplay between the formulation symmetry and the properties of permutation equivariance and invariance in GNNs makes classic GNN architecture incapable of differentiating between symmetric variables.
- Exploring the potential of feature augmentation to address the limitation, and proposing three guiding principles for the construction of augmented features.
- Following these principles, developing an orbit-based feature augmentation scheme, and validating that it can achieve a remarkable improvement of the prediction performance of GNNs for the ILPs with strong symmetry.

## 2 PRELIMINARIES

*Notation*: Unless otherwise specified, scalars are denoted by normal font (i.e., $x$, $A$), vectors are denoted by bold lowercase letters (i.e., $\boldsymbol{x}$), and matrices are represented by bold uppercase letters (i.e., $\boldsymbol{X}$). The $i$-th row and the $j$-th column of a matrix $\boldsymbol{X}$ are denoted by $\boldsymbol{X}_{i,:}$ and $\boldsymbol{X}_{:,j}$, respectively.

### 2.1 ILP AND SYMMETRY

An *integer linear program* (ILP) has a formulation as follows:

$$\min_{\boldsymbol{x}} \{\boldsymbol{c}^\top \boldsymbol{x} | \boldsymbol{A}\boldsymbol{x} \le \boldsymbol{b}, \boldsymbol{x} \in \mathbb{Z}^n\} \tag{1}$$

where $\boldsymbol{x} \in \mathbb{Z}^n$ are integer decision variables, and $\boldsymbol{c} \in \mathbb{R}^n, \boldsymbol{A} \in \mathbb{R}^{m \times n}, \boldsymbol{b} \in \mathbb{R}^m$ are given coefficients. Let $\mathcal{X}$ denote the set of all feasible solutions of (1). A *symmetry* of (1) is a bijection $g : \mathcal{X} \to \mathcal{X}$ such that $\boldsymbol{c}^\top g(\boldsymbol{x}) = \boldsymbol{c}^\top \boldsymbol{x}, \forall \boldsymbol{x} \in \mathcal{X}$. In practice, one often considers symmetries that permute variables (Def. 1) while retaining the description $\boldsymbol{A}\boldsymbol{x} \le \boldsymbol{b}$ and the objective coefficients $\boldsymbol{c}$ invariant. Such symmetries are called *formulation symmetries* (Def. 2). All formulation symmetries of (1) form a group, which is named *symmetry group*.

**Definition 1.** *(Permutation) A permutation over a set $I^n = \{1, \ldots, n\}$ is a bijection $\pi : I^n \to I^n$, such that for every element $i \in I^n$ ($j \in I^n$), there exists a unique element $j \in I^n$ ($i \in I^n$) such that $\pi(i) = j$ ($\pi^{-1}(j) = i$). The set of all $n!$ permutations over $I^n$ is denoted by $S_n$.*

In this work, given a permutation $\pi \in S_n$, when it acts on different objects such as the elements of a vector, the rows and the columns of a matrix, the notation $\pi$ will be added with different superscripts to distinguish them from one another. Specifically, the same permutation $\pi \in S_n$ acting on the top-most $n$ elements of a vector $\boldsymbol{y}$, the top-most $n$ rows and the left-most $n$ columns of a matrix $\boldsymbol{X}$ is denoted by

$$\begin{cases} \pi^v(\boldsymbol{y}) = [\boldsymbol{y}_{\pi(1)}, \ldots, \boldsymbol{y}_{\pi(n)}, \boldsymbol{y}_{n+1}, \ldots]^\top, \\ \pi^r(\boldsymbol{X}) = [\boldsymbol{X}_{\pi(1),:}, \ldots, \boldsymbol{X}_{\pi(n),:}, \boldsymbol{X}_{n+1,:}, \ldots]^\top, \\ \pi^c(\boldsymbol{X}) = [\boldsymbol{X}_{:,\pi(1)}, \ldots, \boldsymbol{X}_{:,\pi(n)}, \boldsymbol{X}_{:,n+1}, \ldots]. \end{cases} \tag{2}$$

**Definition 2.** *(Formulation symmetry) A permutation $\pi \in S_n$ is a formulation symmetry of (1) if there exists a permutation $\sigma \in S_m$ such that*

- $\pi^v(\boldsymbol{c}) = \boldsymbol{c}$,      - $\sigma^v(\boldsymbol{b}) = \boldsymbol{b}$,      - $A_{\sigma(i),\pi(j)} = A_{i,j}, \forall i, j$.

Another important concept in symmetry handling is *orbit* defined in Def. 3, which refers to the set of elements that can be transformed into each other through symmetries. We call two variables are *symmetric* if they correspond to the same orbit.

**Definition 3.** *(Orbit) Let $\mathcal{G}$ be the symmetry group of (1), then the orbit of $i \in I^n$ under $\mathcal{G}$ is a set $\mathcal{O} = \{\pi(i) \mid \forall \pi \in \mathcal{G}\}$. All orbits of $I^n$ under $\mathcal{G}$ form a partitioning of $I^n$, i.e., $\{\mathcal{O}_1, \ldots, \mathcal{O}_K\}$, where $\mathcal{O}_p \cap \mathcal{O}_q = \emptyset, \forall p \neq q \in \{1, \ldots, K\}$ and $\cup_{k=1}^K \mathcal{O}_k = I^n$.*

## 2.2 LEARNING TASKS

In this paper, we consider a classic learning task aimed at developing a model $f_\theta : \mathbb{G} \to \mathbb{R}^n$ to predict an optimal solution for an ILP instance, where $\mathbb{G}$ is the space of the bipartite representation of the ILP, which will be explained in detail later. While there could be multiple optimal solutions for an ILP, this work only focus on predicting just one of them, since many practical applications typically require one solution rather than an exhaustive set. The training of model $f_\theta$ employs supervised learning, utilizing a dataset $\mathcal{D}$ that consists of (input, label) pairs $\{(s_i, \bar{x}_i)\}_{i=1}^N$, where each $s_i$ represents an ILP instance and $\bar{x}_i$ denotes one of its corresponding optimal solutions. The model is trained by minimizing a loss function $\ell(\cdot)$ over all the $N$ instances from the dataset, leading to the optimization problem $\min_\theta \frac{1}{N} \sum_{i=1}^N \ell\left(f_\theta(s_i), \bar{x}_i\right)$.

## 2.3 BIPARTITE REPRESENTATION

In the learning task, an ILP instance $s_i$ is first transformed to an equivalent bipartite graph before being input to the GNN. Without loss of generality, we consider the following bipartite representation. Specifically, an ILP (1) is characterized by a bipartite graph $\{V, W, E\}$, where

- $V = \{v_1, \ldots, v_n\}$ is the set of variable nodes with $v_j \in V$ denoting variable $x_j$. The variable node $v_i$ is associated with feature $h_j^v := c_j \in \mathbb{R}$.

- $W = \{w_1, \ldots, w_m\}$ is the set of constraint nodes with $w_i$ denoting the $i$-th constraint. The constraint node $w_i$ is associated with feature $h_i^w := b_i \in \mathbb{R}$.

- $E = \{e_{ij} \ \forall(i, j) : A_{ij} \neq 0\}$ is the set of edges with $e_{ij}$ denoting that variable $x_j$ appears in the $i$-th constraint. The edge $e_{ij}$ is associated with feature $h_{ij}^e := A_{ij} \in \mathbb{R}$.

An example of the bipartite graph representation is given in Fig. 1. For brevity, we use $\mathcal{A} := \begin{bmatrix} \boldsymbol{H}^e & \boldsymbol{h}^w \\ (\boldsymbol{h}^v)^\top & 0 \end{bmatrix} = \begin{bmatrix} \boldsymbol{A} & \boldsymbol{b} \\ \boldsymbol{c}^\top & 0 \end{bmatrix} \in \mathbb{R}^{(m+1)\times(n+1)}$ to denote the aforementioned bipartite representation.

# 3 ISSUES OCCUR WHEN GNNS MEET FORMULATION SYMMETRY

While GNNs excel at capturing the underlying structure of ILPs, their effectiveness is limited when the ILP problems exhibit specific formulation symmetries. As shown in the example in Fig. 1, the GNN fails to predict the optimal solution for an ILP instance with symmetry.

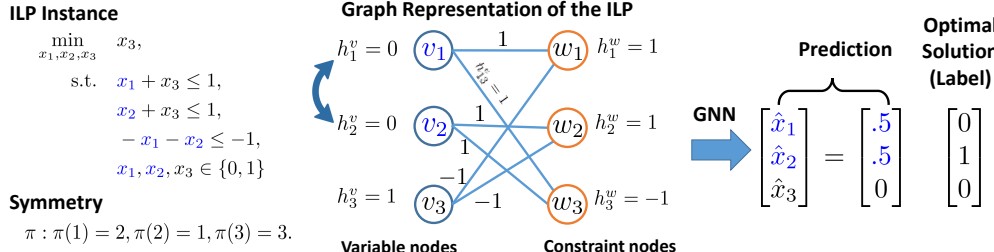

Figure 1: Left: An ILP instance where $x_1$ and $x_2$ are symmetric. Middle: A bipartite representation of the ILP instance. The variable and constraint nodes correspond to their counterparts in the ILP instance, with edges connecting them denoting the coefficients of variables in constraints. Right: The outputs for the symmetric variables are identical due to symmetry, thus GNNs cannot correctly predict the optimal solution.

In the following, we rigorously show that it is the interplay between the inherent properties of GNNs and the formulation symmetry of ILPs that makes the model incapable of distinguishing between symmetric variables and predicting the optimal solutions.

**Assumption 1.** *(Permutation equivariance and invariance) Assume the model $f_\theta$ is equivalent w.r.t. permutations acting on the variable nodes (i.e., $\forall \pi \in S_n, f_\theta(\pi^c(\mathcal{A})) = \pi^v(f_\theta(\mathcal{A}))$) and invariant w.r.t. permutations acting on the constraint nodes (i.e., $\forall \sigma \in S_m, f_\theta(\sigma^r(\mathcal{A})) = f_\theta(\mathcal{A}))$.*

Notice that GNNs naturally satisfy the above assumption. Moreover, when such a model is applied to an ILP instance with formulation symmetry, the elements of the predicted solution corresponding to the same orbit will be identical. Accordingly, the following proposition (see proof in the Appendix A.4) will hold.

**Proposition 1.** *Under Assumption 1, if a permutation $\pi \in S_n$ is a formulation symmetry of (1), then we have $f_\theta(\mathcal{A})_i = f_\theta(\mathcal{A})_{\pi(i)}$. Further, the elements of $f_\theta(\mathcal{A})$ correspond to the same orbit are identical, i.e., $f_\theta(\mathcal{A})_i = f_\theta(\mathcal{A})_j, \forall i, j \in \mathcal{O}, \forall \mathcal{O} \in Orbit(\mathcal{G})$.*

With Proposition 1, it is not difficult to derive the following corollary (see proof in Appendix A.5).

**Corollary 1.** *Under Assumption 1, the model $f_\theta$ cannot always correctly predict the optimal solution of an ILP instance with formulation symmetries.*

## 4 METHODOLOGY

The analysis in the previous section raises a key question: How can we improve the ability of GNNs to solve ILPs with formulation symmetry? A promising approach is to augment the input features of the GNN, enabling it to differentiate between symmetric variables. While a similar idea was adopted in (Chen et al., 2022), where random features were added into the bipartite graph representation to distinguish what they refer to as 'foldable' ILP instances, the approach did not exploit the underlying symmetry properties, leading to suboptimal performance on ILPs with strong symmetries. In this section, we first establish three guiding principles for feature augmentation and, based on these principles, propose an orbit-based feature augmentation scheme to handle symmetries more effectively.

### 4.1 PRINCIPLES FOR CONSTRUCTING AUGMENTED FEATURES

Motivated by the existing symmetry-breaking methods (Chen et al., 2022; Xie & Smidt, 2024; Lawrence et al., 2024) that address symmetry by introducing augmented features into the data, we tackle the indistinguishability issue in ILPs by incorporating random features into the bipartite graph representation. Specifically, let $z \in \mathbb{R}^n$ be an augmented feature sampled from a space $\mathbb{V} \subseteq \mathbb{R}^n$, and it is assigned to the $n$ variable nodes. For brevity, let $\tilde{\mathcal{A}} = \begin{bmatrix} \mathcal{A} \\ z^\top & 0 \end{bmatrix} = \begin{bmatrix} A & b \\ c^\top & 0 \\ z^\top & 0 \end{bmatrix} \in \mathbb{R}^{(m+2)\times(n+1)}$

be the bipartite graph representation incorporating $z$. In contrast to existing methods that add augmented features to all nodes in the graph, we focus exclusively on the variable nodes. This strategy targets the symmetries inherent in the variables, which are sufficient to differentiate the outputs during solution prediction, while disregarding the constraint symmetries. By doing so, it simplifies the symmetry groups that need to be processed, improving computational efficiency.

There are three principles the augmented feature $z$ should follow:

- (distinguishability) there exists a function $f_\theta$ such that $f_\theta(\tilde{\mathcal{A}})_i \neq f_\theta(\tilde{\mathcal{A}})_j, \forall i \neq j \in \mathcal{O}, \forall \mathcal{O} \in Orbits(\mathcal{A}, \mathcal{G})$.
- (augmentation parsimony) The cardinality of the augmented feature space $\mathbb{V}$ should be as small as possible.
- (isomorphic consistency) If $(\mathcal{A}, \bar{x})$ and $(\mathcal{A}', \bar{x}')$ are two training samples with isomorphic instances $\mathcal{A}$ and $\mathcal{A}'$ (i.e., $\exists \pi \in S_n, \sigma \in S_m$ such that $\pi^c(\sigma^r(\mathcal{A})) = \mathcal{A}'$), then $\pi^v(z) = z' \implies \pi^v(\bar{x}) = \bar{x}'$.

The first principle, *distinguishability*, is a necessity, which enables GNNs to output different values for symmetric variables. The simplest way to realize it is by assigning distinct features to variables

in the same orbit, i.e., $z_i \neq z_j, \forall i \neq j \in \mathcal{O}$. The second principle, *augmentation parsimony*, plays a crucial role in the model's training process. By using a small cardinality of the augmented feature space, this guiding principle prevents the model from being overwhelmed by excessive irrelevant information that could slow down the learning of correct correlations. This enhances training efficiency, as fewer features require less computational effort to learn and stabilize the model, leading to faster convergence and better overall performance. Note that the core ideas underlying these two principles are drawn from existing works (Xie & Smidt, 2024; Lawrence et al., 2024; Morris et al., 2024) and have been adapted to fit our augmentation scheme.

The last principle, *isomorphic consistency*, enforces that the labels of isomorphic inputs should remain isomorphic as well. This principle stems from the permutation equivariance/invariance of the ground truth function, with further details provided in Appendix A.2. Samples that fail to meet this criterion are termed *conflict* or *inconsistent* samples, which can negatively impact the GNN's training. Proposition 2 (see proof in the Appendix A.6) reveals that *conflict* samples will lead to a higher loss and should be avoided in constructing the training data.

**Proposition 2.** *If the augmented features doesn't satisfy the principle of isomorphic consistency, then the minimal loss can not be* 0.

## 4.2 ORBIT-BASED FEATURE AUGMENTATION

Following the aforementioned three guiding principles, we develop a novel feature augmentation scheme that harnesses the formulation symmetry of ILPs in constructing the augmented features.

First, the principle of distinguishability can be easily achieved; the simplest approach is to assign a unique augmented feature to each variable node. This is equivalent to sampling $\{z_i\}$ from the set $\{1, \ldots, n\}$ without replacement. However, the cardinality of the augmented feature space by such an approach is $|\mathbb{V}| = n!$, which is too large to ensure a good augmentation parsimony.

Additionally, the isomorphic consistency property stipulates that the augmented features $z$ and $z'$ for any two training samples $(\mathcal{A}, \bar{x}), (\mathcal{A}', \bar{x}')$ with isomorphic instances should satisfy $\pi^v(z) = z' \implies \pi^v(\bar{x}) = \bar{x}'$. This relation is imposed on not only the augmented feature but also the label. For ease of illustration, here we introduce the construction of augmented features first and leave the analysis on how isomorphic consistency is handled to Sec. 4.3.

---

**Algorithm 1** Orbit-based feature augmentation

1: **Input:** ILP instance $\mathcal{A}$ with orbits $\{\mathcal{O}_1, \ldots, \mathcal{O}_K\}$ .
2: **Procedure:**
3: Initialize $z \leftarrow \mathbf{0}$
4: **for** $k \in \{1, \ldots, K\} : |\mathcal{O}_k| \geq 2$ **do**        ▷ nontrivial orbits
5:     $\mathcal{C} \leftarrow \{1, \ldots, |\mathcal{O}_k|\}$
6:     **for** $i \in \mathcal{O}_k$ **do**
7:         $z_i \sim \text{Uniform}(\mathcal{C})$                  ▷ sampling
8:         $\mathcal{C} = \mathcal{C} \setminus \{z_i\}$                ▷ without replacement
9:     **end for**
10: **end for**
11: **Output:** augmented feature $z$

---

Further, in accordance with the principle of augmentation parsimony, the augmented features with a smaller cardinality of their associated space $|\mathbb{V}|$ are preferable. Intuitively, $z_i$ should be sampled over a set smaller than $\{1, \ldots, n\}$ while maintaining the distinguishability. For ILPs with symmetry, one can find that sampling over the orbits can achieve the two targets at once. To this end, we propose an orbit-based feature augmentation approach as follows. Specifically, consider an ILP instance $\mathcal{A}$ characterized by its orbits $\{\mathcal{O}_1, \ldots, \mathcal{O}_K\}$. For any orbit $\mathcal{O}_k \in \{\mathcal{O}_1, \ldots, \mathcal{O}_K\}$ with $|\mathcal{O}_k| \geq 2$, the augmented feature $\{z_i\}, i \in \mathcal{O}_k$ are uniformly sampled from a discrete set $\mathcal{C} = \{1, 2, \ldots, |\mathcal{O}_k|\}$ without replacement. For trivial orbits that contain only a single element, we assign zeros as the corresponding augmented features. The proposed scheme is detailed in Algorithm 1 and is referred to as **Orbit** in the remainder of this work.

By leveraging the structural symmetry, the proposed **Orbit** effectively reduces the cardinality of the augmented feature space. In many categories of ILPs, there usually exist certain connections between different orbits, which can be further exploited to reduce the cardinality. Specifically, assume an ILP instance has $p \leq K$ orbits $\mathcal{O}_1, \ldots, \mathcal{O}_p$ with the same cardinality $c$. The elements of these orbits form a matrix $\boldsymbol{O} = \begin{bmatrix} o_{11} & \ldots & o_{1c} \\ & \ldots & \\ o_{p1} & \ldots & o_{pc} \end{bmatrix}$, where $o_{ij} \in \mathcal{O}_i, i \in \mathcal{P} = \{1, \ldots, p\}$.

The formulation symmetry of the instance necessitates that all elements in each column of $\boldsymbol{O}$ be

treated as an integrated unit under any permutations defined by the symmetry group. That is, $\forall \pi \in \mathcal{G}, \pi(o_{ij}) = o_{ik} \Leftrightarrow \pi(o_{i'j}) = o_{i'k}, \forall i, i' \in \mathcal{P}, k \in \{1, \ldots, c\}$. For such an instance, the augmented features added to the variables corresponding to the same column of $\boldsymbol{O}$ can be identical. Accordingly, it suffices to sample augmented features for the variables in one orbit and assign the same features to the corresponding variables in the other orbits, e.g., $o_{1j} \leftarrow \bar{m}_j, \forall j \in \{1, \ldots, c\} \implies o_{ij} \leftarrow \bar{m}_j, \forall i \in \{1, \ldots, p\}, j \in \{1, \ldots, c\}$. As shown in the example in Appendix A.1, this updated scheme, named **Orbit+**, further employs the additional connections among the orbits in constructing the augmented features, achieving a smaller $|\mathbb{V}|$ with enhanced augmentation parsimony.

## 4.3 ANALYSIS

In this section, our proposed orbit-based feature augmentation schemes and two other existing schemes are analyzed with our proposed three principles in Sec. 4.1. Specifically, we will evaluate whether the distinguishability and isomorphic consistency are satisfied, as well as assess the cardinalities of their respective augmented feature spaces. Before proceeding with the analysis, let's first introduce two existing feature augmentation schemes.

**Random noise from a uniform distribution (Uniform)**   Chen et al. (2022) noticed the lack of expressive power of GNNs to distinguish some ILP instances (called as "foldable"), and proposed to introduce random noise from uniform distribution to both variable and constraint nodes. Here we consider the case where random noise is added to variable nodes only, as it is sufficient to meet the distinguishability.

**Positional IDs (Position)**   The second scheme is adding positional IDs to the variable nodes, ensuring each variable node is associated with a distinct ID. Han et al. (2023) adopted such a trick in their feature designs, while without discussing insights and necessity.

### 4.3.1 ON THE PRINCIPLES OF DISTINGUISHABILITY AND ISOMORPHIC CONSISTENCY

The condition outlined in the principle of distinguishability is straightforward to satisfy, provided that the variable nodes within the same orbit are associated with distinct augmented features. It's easy to verify that the Position, Orbit, and Orbit+ schemes all strictly assign distinct augmented features to variable nodes in the same orbit, while the Uniform scheme does so with probability 1. Therefore, all these four schemes meet the principle of distinguishability.

Unlike the principle of distinguishability, the principle of isomorphic consistency imposes a more complex condition, as it applies to both the input instances and the output labels. As demonstrated in Appendix A.3, sampling strategies can violate this principle, resulting in situations where $\pi^v(\boldsymbol{z}) = \boldsymbol{z}'$ but $\pi^v(\bar{\boldsymbol{x}}) \neq \bar{\boldsymbol{x}}'$. To address this issue, there are two potential approaches: one is resampling when $\pi^v(\boldsymbol{z}) = \boldsymbol{z}'$, while the other replaces $\bar{\boldsymbol{x}}$ and $\bar{\boldsymbol{x}}'$ with alternative optimal solutions $\bar{\boldsymbol{y}}$ and $\bar{\boldsymbol{y}}'$, ensuring that $\pi^v(\bar{\boldsymbol{y}}) = \bar{\boldsymbol{y}}'$. In this paper, we adopt the second approach, as the first one relies on label-dependent reject sampling, which is infeasible during the testing phase when labels are unavailable. Specifically, we utilize the SymILO framework proposed by Chen et al. (2024), which supports dynamically adjusting the labels of the training samples. It jointly optimizes the transformation of solutions and the model parameters, aiming to minimize the prediction error. Since SymILO does not directly operate on augmented features, we apply it uniformly across all methods in Section 5 to alleviate the impacts of violations of the principle of isomorphic consistency.

### 4.3.2 ON THE PRINCIPLE OF AUGMENTATION PARSIMONY

Among the four augmentation schemes, the augmented feature of Uniform is sampled from a continuous uniform distribution. Therefore, the cardinality of its augmented feature space can be infinite, i.e., $c_u = +\infty$. For the Position scheme, its augmented features are uniformly sampled from a discrete distribution from the set $\{1, \ldots, n\}$. Accordingly, the cardinality of its feature space is $c_p = n!$. In comparison, for our proposed Orbit scheme described in Sec. 4.2, the augmented feature $\{z_i\}$ associated with each orbit $\mathcal{O}_k \in Orbits(\mathcal{A})$ is sampled from the set $\{1, \ldots, |\mathcal{O}|_k\}$. As a result, the cardinality of our proposed Orbit scheme is $c_o = (|\mathcal{O}_1|!) \cdot (|\mathcal{O}_2|!) \cdot \ldots \cdot (|\mathcal{O}_K|!)$, where $\sum_{i=1}^{K} |\mathcal{Q}_i| \leq n$. Further, when the formulation symmetry imposes additional connections

among $1 < p \leq K$ orbits, the augmented feature only needs to be sampled for one orbit and can be reused for the other $(p-1)$ orbits. Without loss of generality, assume these $p$ orbits are composed by $\mathcal{O}_1, \mathcal{O}_2, \ldots, \mathcal{O}_p$ with $|\mathcal{O}_1| = |\mathcal{O}_2| = \cdots = |\mathcal{O}_p|$. Correspondingly, the cardinality of the augmented feature space of Orbit+ is $c_{o+} = (|\mathcal{O}_p|!) \cdot \cdots \cdot (|\mathcal{O}_K|!)$. It is not difficult to verify that $c_{o+} < c_o < c_p < c_u$. Correspondingly, our proposed orbit-based augmentation schemes achieve a better augmentation parsimony.

## 5 EXPERIMENTS

In this section, we present numerical experiments to validate the effectiveness of the proposed approaches. The source code is available at https://github.com/NetSysOpt/GNNs_Sym_ILPs.

### 5.1 DATASET

We evaluate our proposed approach using three ILP benchmark problems that exhibit significant symmetry. The descriptions of these benchmarks are as follows:

**BPP:** The bin packing problem (BPP) is a well-known practical problem where items must be placed into bins without exceeding capacity limits. The objective is to minimize the total number of bins used. We generate 500 instances, each with 20 items, following the generation strategies outlined by Schwerin & Wäscher (1997). These instances include 420 variables and 40 constraints, with an average of 14 orbits, and orbit cardinalities reaching up to 140.

**BIP:** The balanced item placement problem (BIP) also involves assigning items to bins. However, unlike bin packing, the goal is to balance resource usage across bins. We use 300 instances from the ML4CO competition benchmarks (Gasse et al., 2022). These instances feature 1,083 variables and 195 constraints, with an average of 100 orbits.

**SMSP:** The steel mill slab design problem (SMSP) is a variant of the cutting stock problem, where customer orders are assigned to slabs under color constraints, with the aim of minimizing total waste. We use 380 instances from (Schaus et al., 2011), which range between 22,000 and 24,000 variables and nearly 10,000 constraints. On average, these instances have 110 orbits, with orbit cardinalities varying between 111 and 1,000.

In our experiments, 60% of the instances are used for training, and the remaining 40% are reserved for validation. Since these datasets include only the problem instances, we also gather corresponding solutions. Due to the complexity of the constraints, obtaining optimal solutions for every instance is not computationally feasible. Instead, we run the ILP solver SCIP (Gamrath et al., 2020) for 3,600 seconds on each instance and store the best solution found. The average numbers of variables, constraints, as well as orbits of each benchmark problem, are summarized in Table 1.

Table 1: Statistics about the datasets.

| Problem | Avg. number | | |
|---------|------|------|--------|
|         | Var. | Cons. | Orbits |
| BPP  | 420    | 40    | 14  |
| BIP  | 1,083  | 195   | 100 |
| SMSP | 23,000 | 1,000 | 110 |

### 5.2 BASELINES AND THE PROPOSED METHODS

We consider three baselines (**No-Aug**, **Uniform** and **Position**) which employ different feature augmentation strategies, and compare them to our proposed methods (**Orbit** and **Orbit+**).

**No-Aug:** This is the baseline where no feature augmentation is adopted. To align with other strategies, the augmented features $z$ are set as zeros for all variables. **Uniform:** As described in Sec. 4.3, this baseline samples each element of $z$ individually from a uniform distribution to distinguish symmetric variables, i.e., $z_i \sim \mathcal{U}(0,1), \forall i \in [n]$. **Position:** As described in Sec. 4.3, this baseline assigns unique integer numbers to the elements of $z$ to distinguish between different variables by their positions. Specifically, the augmented features $z_i, \forall i \in [n]$ can be uniformly sampled from $\{1, \ldots, n\}$ without replacement. **Orbit:** This is our proposed augmentation scheme outlined in Algorithm 1, which utilizes the structural information from orbits and adds augmented features within each orbit individually. **Orbit +:** This is an enhanced version of the orbit-based feature augmen-

tation scheme mentioned in Section 4.2, which exquisitely assigns the same augmented features to multiple orbits for certain types of symmetries.

## 5.3 EVALUATION METRICS

To evaluate the prediction performance of models trained with different augmented features, the *Top-m%* error proposed by Chen et al. (2024) is used as the evaluation metric, which takes into account the impact of the formulation symmetry on the solutions.

**Top-$m\%$ error:** It is based on the $\ell_1$-distance between a rounded prediction and its closest symmetric solution. Given the label $y$ of a instance and a prediction $\hat{y}$, the equivalent solution of $y$ closest to $\hat{y}$ is defined as $\tilde{y} = \pi'(y)$, where $\pi' = \arg\min_\pi \|\hat{y} - \pi(y)\|$. Based on this observation, the Top-$m\%$ error is defined as:

$$\mathcal{E}(m) = \sum_{i \in M} |\text{Round}(\hat{y}_i) - \tilde{y}_i|, \qquad (3)$$

where $M$ is the index set of the top $m\%$ variables with the smallest values of $|\text{Round}(\hat{y}_j) - \hat{y}_j|, \forall j$. This error measures the minimum $\ell_1$-distance between the prediction and all equivalent solutions of the label. Compared to the $\ell_1$ distance $\sum_{i \in M} |\text{Round}(\hat{y}_i) - y_i|$ that ignores the effect of multiple equivalent solutions caused by formulation symmetry, the metric adopted characterizes the distance between a prediction and a feasible solution more accurately. With the standard $\ell_1$-distance, the error would be non-zero when $\text{Round}(\hat{y}) \neq y$, whereas it would be reduced to 0 with (3), as long as there exists a $\tilde{y} = \pi'(y)$ and its element $\tilde{y}_i$ matches $\text{Round}(\hat{y}_i)$.

## 5.4 MODEL AND TRAINING SETTINGS

The model architecture follows Han et al. (2023), where four half-layer graph convolutions are used to extract hidden features and another two-layer perceptron is used to make the final prediction. We also follow most of their settings for the initial features used in the bipartite representation of ILP instances. The only difference is we omit the "pos_emb" feature as its role is the same as the "Pos" augmentation strategy in our baselines.

In the training configuration, we utilize the Adam optimizer with a learning rate of 0.0001 and a batch size of 8. All models are trained for 100 epochs, with the parameters corresponding to the lowest validation loss preserved for subsequent evaluation. Since all augmented features are randomly generated, multiple samples should be drawn for each training instance to mitigate overfitting. Accordingly, we sample 8 times for each training instance, while only a single sample is taken for each instance in the test set. The symmetry detection is conducted with the well-developed tool Bliss, and more details are shown in Appendix A.8.

## 5.5 MAIN RESULTS

In this section, we present the numerical results comparing different augmented features. As shown in Table 2, among the three baselines, the Top-$m\%$ errors attained by Uniform and those by Position are much smaller than those by No-Aug. This is natural since the last one does not employ any feature augmentation. The large error of No-Aug demonstrates the necessity of addressing the issue that symmetric variables can not be distinguished. Meanwhile, the performance improvement brought by Uniform and Poisition validates the effectiveness of feature augmentation. Moreover, Position has a smaller cardinality of augmented feature space, thus its performance is generally better than that of Uniform.

Compared with the three baselines, our proposed orbit-based feature augmentation methods bring a notable reduction of the prediction errors. Notice that the reduction of loss from Position to Orbit is more distinct than that from Uniform to Position. This is because both Uniform and Position mainly consider enhancing the distinguishability in the feature augmentation but overlooking the inherent formulation symmetry of these problems. In contrast, our proposed Orbit-based methods are symmetry-aware, where the associated augmented features are constructed explicitly based on the symmetry groups. As a result, the cardinality of the augmented feature space of our proposed Orbit and that of Orbit+ can be smaller, as analyzed in Sec. 4.3.2. The remarkable improvement of the

prediction accuracy of Orbit and Orbit+ demonstrates the superiority of the symmetry-aware feature augmentation and validates the effectiveness of our proposed guiding principles for constructing augmented features. Besides, one can observe that Orbit+ generally attains a better Top-$m\%$ error performance than Orbit. This is in accordance with our expectation since Orbit+ leverages more symmetry information in constructing the augmented features and further reduces the cardinality of the feature space.

Table 2: Top-$m\%$ errors ($\downarrow$) of different feature augmentation schemes.

| Methods | BPP | | | | BIP | | | | SMSP | | | |
|---|---|---|---|---|---|---|---|---|---|---|---|---|
| | 30% | 50% | 70% | 90% | 30% | 50% | 70% | 90% | 30% | 50% | 70% | 90% |
| No-Aug | 3.0 | 4.8 | 6.6 | 9.5 | 30.4 | 50.6 | 71.0 | 91.1 | 34.0 | 57.7 | 83.0 | 113.9 |
| Uniform | 0.0 | 0.4 | 2.4 | 6.4 | 4.8 | 15.9 | 44.9 | 80.2 | 18.5 | 34.7 | 53.3 | 80.0 |
| Position | 0.0 | 0.0 | 1.3 | 5.6 | 4.5 | 13.4 | 45.6 | 81.6 | 19.3 | 35.2 | 53.4 | 79.8 |
| **Orbit** | 0.0 | 0.0 | 1.3 | **4.3** | 3.6 | 8.6 | **39.0** | **75.7** | 0.0 | 1.5 | 17.9 | 51.1 |
| **Orbit +** | 0.0 | 0.0 | **0.9** | **4.3** | **3.2** | **5.5** | 39.4 | 79.2 | 0.0 | **1.0** | **14.9** | **50.3** |

In Fig. 2, the validation losses versus epochs of these baseline feature augmentation methods as well as our proposed Orbit and Orbit+ are presented. It is evident that the attained validation loss after convergence satisfies Orbit+ < Orbit < Position < Uniform. This is consistent with the trend of the Top-$m\%$ errors in Table 1. Additionally, one can observe that the validation losses of Orbit and Orbit+ drop more quickly than those of the baselines. Specifically, Orbit+ merely takes around 20 epochs to reach the smallest loss over the BPP and BIP datasets, while Uniform and Position take around $30 \sim 40$ epochs. The phenomenon is not surprising, since a smaller cardinality of augmented space has the potential to achieve better training efficiency as analyzed in Section 4.1. The results in Fig. 2 and Table 2 confirm that our proposed orbit-based feature augmentation not only provides more accurate solution predictions but also enhances the training efficiency of the learning model, offering a competitive approach for solving ILPs with symmetries. Besides the main results, supplementary numerical results are available in Appendix A.7.

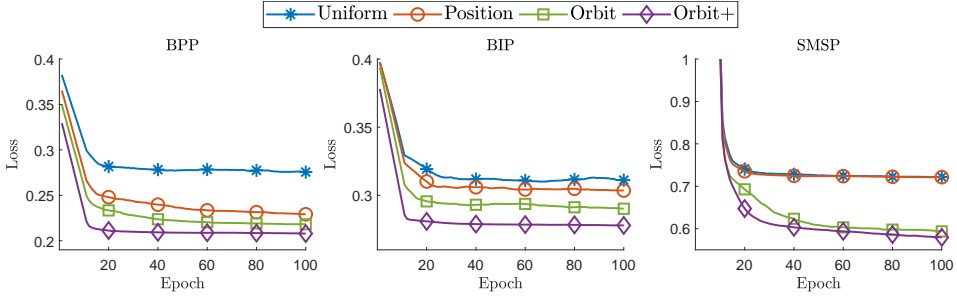

Figure 2: Validation losses of different schemes.

## 6 CONCLUSION AND LIMITATION

In this work, we demonstrated that the interaction between the formulation symmetry of ILPs and the permutation invariance and equivariance properties of GNNs limits the ability of classic GNN architectures to distinguish between symmetric variables. Exploring the potential of feature augmentation to address this limitation, we proposed three guiding principles for constructing the augmented features. Based on these principles, we developed a new orbit-based feature augmentation scheme, which can distinctively enhance the prediction performance of GNNs for ILPs with symmetry. There are several limitations of our orbit-based augmentation scheme, which also present opportunities for future research. First, our approach is specifically tailored for ILPs with formulation symmetry, and it is currently unknown whether it can be effectively applied to problems of other classes. Second, the principle of isomorphic consistency, which underpins our method, is primarily applicable to supervised learning tasks where multiple label choices exist. These limitations highlight areas for further exploration and potential extension of our approach.

## ACKNOWLEDGMENTS

This work was supported by the National Key R&D Program of China under grant 2022YFA1003900. Akang Wang also acknowledges support from the National Natural Science Foundation of China (Grant No. 12301416), the Guangdong Basic and Applied Basic Research Foundation (Grant No. 2024A1515010306), the Shenzhen Science and Technology Program (Grant No. RCBS20221008093309021), and the Longgang District Special Funds for Science and Technology Innovation (LGKCSDPT2023002). Ruoyu Sun also acknowledges support from the Hetao Shenzhen-Hong Kong Science and Technology Innovation Cooperation Zone Project (No. HZQSWS-KCCYB-2024016), the University Development Fund (UDF01001491) at the Chinese University of Hong Kong, Shenzhen, the Guangdong Provincial Key Laboratory of Mathematical Foundations for Artificial Intelligence (2023B1212010001), and the Guangdong Major Project of Basic and Applied Basic Research (2023B0303000001). Qian Li also acknowledges support from Hetao Shenzhen-Hong Kong Science and Technology Innovation Cooperation Zone Project (No.HZQSWS-KCCYB-2024016). Tsung-Hui Chang acknowledges support from the Shenzhen Science and Technology Program (Grant No. ZDSYS20230626091302006).

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

# A  APPENDIX

## A.1  EXAMPLE OF CONNECTIONS BETWEEN ORBITS

**Example 1.** *Consider a bin-packing problem with 3 items of sizes $\{1, 3, 5\}$ and up to 3 bins, each with a capacity of 5. The objective is to minimize the number of bins used while ensuring the total size of items in each bin does not exceed its capacity. Let $x_{ij}$ be a binary variable where $x_{ij} = 1$ if item $i$ is placed in bin $j$, and $y_j$ be a binary variable where $y_j = 1$ if bin $j$ is used. Then this problem can be formulated as:*

$$\min_{x_{ij}, y_j} \quad y_1 + y_2 + y_3$$

$$\text{s.t.} \quad x_{i1} + x_{i2} + x_{i3} = 1, i = 1, 2, 3 \tag{4a}$$

$$1x_{1j} + 3x_{2j} + 5x_{3j} \leq 5y_j, j = 1, 2, 3 \tag{4b}$$

$$x_{ij}, y_j \in \{0, 1\} \tag{4c}$$

The formulation symmetries of this problem are arbitrary permutations acting on the indices $j$, namely, permutations acting on the columns of matrix $\boldsymbol{X} = \begin{bmatrix} x_{11} & x_{12} & x_{13} \\ x_{21} & x_{22} & x_{23} \\ x_{31} & x_{32} & x_{33} \\ y_1 & y_2 & y_3 \end{bmatrix}$. There are 4 orbits, each corresponding to a row of $\boldsymbol{X}$. In the basic orbit-based feature augmentation approach described in Algorithm 1, the augmented features added to the variables in each column of $\boldsymbol{X}$ can be distinct. However, due to the symmetries inherent in this problem, each column should be treated as an indivisible unit. Consequently, the features added to different variables within the same column should be identical. Specifically, following Orbit+, we only need to sample $z_k, k = 1, 2, 3$ from $\{1, 2, 3\}$ for the variables in the first row of $\boldsymbol{X}$ and applies $\boldsymbol{z}_1 = [z_1, z_2, z_3]^\top$ to the variables in the other rows. The attained augmented feature associated with the variables $\boldsymbol{x} = [\boldsymbol{X}_{1,:}, \boldsymbol{X}_{2,:}, \boldsymbol{X}_{3,:}, \boldsymbol{X}_{4,:}]^\top$ will be $\boldsymbol{z}_{o+} = [\boldsymbol{z}_1^\top, \boldsymbol{z}_1^\top, \boldsymbol{z}_1^\top, \boldsymbol{z}_1^\top]^\top$. In comparison, following Orbit, the attained augmented feature $\boldsymbol{z}_o = [\boldsymbol{z}_1^\top, \boldsymbol{z}_2^\top, \boldsymbol{z}_3^\top, \boldsymbol{z}_4^\top]^\top$, where the elements of each $\boldsymbol{z}_i, i = 1, \ldots, 4$ are individually sampled from $\{1, 2, 3\}$. Obviously, the cardinality of the space of $\boldsymbol{z}_{o+}$ is smaller.

## A.2  MORE EXPLANATION OF ISOMORPHIC CONSISTENCY IN SECTION 4.1

Let $f^*$ denote the ground truth function for the learning task described in Section 2.2. A key motivation for using GNNs to approximate $f^*$ is that $f^*$ exhibits permutation invariance and permutation equivariance, i.e., $\forall \pi \in S_n, f_\theta(\pi^c(\cdot)) = \pi^v(f_\theta(\cdot))$, and $\forall \sigma \in S_m, f_\theta(\sigma^r(\cdot)) = f_\theta(\cdot)$. However, introducing additional features may disrupt these properties.

To prevent this from happening, we examine the conditions that need to be satisfied under these two properties. Specifically, let $(\tilde{\mathcal{A}}, \bar{\boldsymbol{x}})$ and $(\tilde{\mathcal{A}}', \bar{\boldsymbol{x}}')$ be two training samples with $\pi^c(\sigma^r(\tilde{\mathcal{A}})) = \tilde{\mathcal{A}}'$ for some $\pi \in S_n$ and $\sigma \in S_m$ (i.e., $\pi^c(\sigma^r(\mathcal{A})) = \mathcal{A}'$ and $\pi^v(\boldsymbol{z}) = \boldsymbol{z}'$). As points on the ground truth function $f^*$, these two training samples are equivalent to $f^*(\tilde{\mathcal{A}}) = \bar{\boldsymbol{x}}$ and $f^*(\tilde{\mathcal{A}}') = \bar{\boldsymbol{x}}'$. Due to the permutation equivariance and invariance of $f^*$, we obtain

$$\bar{\boldsymbol{x}}' = f^*(\tilde{\mathcal{A}}') = f^*(\pi^c(\sigma^r(\tilde{\mathcal{A}}))) = \pi^v(f^*(\tilde{\mathcal{A}})) = \pi^v(\bar{\boldsymbol{x}}).$$

In summary, if $\exists \pi \in S_n$ and $\sigma \in S_m$ such that $\pi^c(\sigma^r(\mathcal{A})) = \mathcal{A}'$, then $\pi^v(\boldsymbol{z}) = \boldsymbol{z}' \implies \pi^v(\boldsymbol{x}) = \boldsymbol{x}'$ should hold.

## A.3  EXAMPLES FOR THE PRINCIPLE OF ISOMORPHIC CONSISTENCY

Consider two isomorphic instances as follows:

$$\min_{x_1, x_2, x_3} \quad x_1 + x_2 + 3x_3$$

$$\text{s.t.} \quad x_1 + x_2 = 1 \tag{5}$$

$$x_1, x_2, x_3 \in \{0, 1\}$$

$$\min_{x_1,x_2,x_3} \quad 3x_1 + x_2 + x_3$$
$$\text{s.t.} \quad x_2 + x_3 = 1 \tag{6}$$
$$x_1, x_2, x_3 \in \{0, 1\}$$

Let $\mathcal{A}$ represent the first instance and $\mathcal{A}'$ represent the second one, then $\pi^c(\sigma^r(\mathcal{A})) = \mathcal{A}'$ with $\pi : \pi(1) = 3, \pi(3) = 1, \pi(2) = 2$ and $\sigma : \sigma(1) = 1$. The label of $\mathcal{A}$ is $\bar{\boldsymbol{x}} = [0, 1, 0]^\top$ and that of $\mathcal{A}'$ is $\bar{\boldsymbol{x}}' = [0, 0, 1]^\top$. Assume the augmented features are $\boldsymbol{z} = [1, 2, 0]^\top$ for $\mathcal{A}$ and $\boldsymbol{z}' = [0, 2, 1]^\top$ for $\mathcal{A}'$. Correspondingly, $\tilde{\mathcal{A}} = \begin{bmatrix} 1 & 1 & 0 & 1 \\ 1 & 1 & 3 & 0 \\ 1 & 2 & 0 & 0 \end{bmatrix}$ and $\tilde{\mathcal{A}}' = \begin{bmatrix} 0 & 1 & 1 & 1 \\ 3 & 1 & 1 & 0 \\ 0 & 2 & 1 & 0 \end{bmatrix}$. It can be easily verified that the property of distinguishability is satisfied by both $\tilde{\mathcal{A}}$ and $\tilde{\mathcal{A}}'$. However, it is evident that $\pi^v(\boldsymbol{z}) = \boldsymbol{z}'$ but $\pi^v(\bar{\boldsymbol{x}}) \neq \bar{\boldsymbol{x}}'$, so that the property of isomorphic consistency is violated. From the perspective of GNN, $\mathcal{A}$ and $\mathcal{A}'$ are equivalent since $\pi^v(\boldsymbol{z}) = \boldsymbol{z}'$. However, the two instances $(\tilde{\mathcal{A}}, \bar{\boldsymbol{x}})$ and $(\tilde{\mathcal{A}}', \bar{\boldsymbol{x}}')$ have different labels as $\pi^v(\bar{\boldsymbol{x}}) \neq \bar{\boldsymbol{x}}'$. This will result in 'conflict' samples, which will mislead the prediction of GNNs to diverge from the correct solutions.

## A.4 Proof of Proposition 1

Without loss of generality, assume the loss function $\ell(\cdot) \geq 0$ is permutation-invariant (i.e., $\ell(\pi^v(\boldsymbol{a}), \pi^v(\boldsymbol{b})) = \ell(a, b)$) and possesses the identity property (i.e., $\ell(\boldsymbol{a}, \boldsymbol{b}) = 0 \iff \boldsymbol{a} = \boldsymbol{b}$, examples of such loss functions include mean-squared loss and cross-entropy loss.).

**Proof:** Since $\pi \in S_n$ is a formulation symmetry, so there exists $\sigma \in S_m$ such that $\pi^c(\sigma^r(\mathcal{A})) = \mathcal{A}$. Besides, $f_\theta$ has permutation equivariance and invariance properties, so we have $f_\theta(\mathcal{A}) = f_\theta(\pi^c(\sigma^r(\mathcal{A}))) = \pi^v(f_\theta(\sigma^r(\mathcal{A}))) = \pi^v(f_\theta(\mathcal{A}))$, i.e., $f_\theta(\mathcal{A})_i = f_\theta(\mathcal{A})_{\pi(i)}$. Further, $\forall i, j \in \mathcal{O}$, there exists $\pi \in \mathcal{G}$ such that $j = \pi(i)$, so $f_\theta(\mathcal{A})_i = f_\theta(\mathcal{A})_j$.

## A.5 Proof of Corollary 1

**Proof:** (counter example) Consider an ILP $s = \min\{x_3 | x_1 + x_2 + x_3 = 1, x_1, x_2, x_3 \in \{0, 1\}\}$, it has a permutation symmetry ($\pi(1) = 2, \pi(2) = 1, \pi(3) = 3$) and two optimal solutions $(0, 1), (1, 0)$. Since indices 1 and 2 can be permuted, thus 1 and 2 are in the same orbit, and $f_\theta(s)_1 = f_\theta(s)_2$. However, both the two optimal solutions $(0, 1, 0)$ and $(1, 0, 0)$ have distinct values for $x_1$ and $x_2$. Therefore, $f_\theta$ cannot predict any optimal solution of $s$.

## A.6 Proof of Proposition 2

**Proof:** Consider two samples $(\mathcal{A}, \bar{\boldsymbol{x}})$ and $(\mathcal{A}', \bar{\boldsymbol{x}}')$ where $\exists \pi \in S_n$, such that $\pi^c(\mathcal{A}) = \mathcal{A}'$, and $\pi^v(\boldsymbol{z}) = \boldsymbol{z}'$. Accordingly, $\pi^c(\tilde{\mathcal{A}}) = \tilde{\mathcal{A}}'$. After adding augmented features, the total loss of these two samples is $L = \ell(f_\theta(\tilde{\mathcal{A}}), \bar{\boldsymbol{x}}) + \ell(f_\theta(\tilde{\mathcal{A}}'), \bar{\boldsymbol{x}}')$. For the second term, it will hold true that $\ell(f_\theta(\tilde{\mathcal{A}}'), \bar{\boldsymbol{x}}') = \ell(f_\theta(\pi^c(\tilde{\mathcal{A}})), \bar{\boldsymbol{x}}') = \ell(\pi^v(f_\theta(\tilde{\mathcal{A}})), \bar{\boldsymbol{x}}') = \ell(f_\theta(\tilde{\mathcal{A}}), (\pi^v)^{-1}(\bar{\boldsymbol{x}}'))$, where the first equality holds since $\mathcal{A}$ and $\mathcal{A}'$ are two isomorphic instances. The second equality holds as $f_\theta$ is permutation-invariant, and the third equality holds as $\ell(\cdot)$ is permutation-invariant and $\pi$ is a bijection. If the principle of isomorphic consistency is violated, then $\pi^v(\bar{\boldsymbol{x}}) \neq \bar{\boldsymbol{x}}'$, namely $\bar{\boldsymbol{x}} \neq (\pi^v)^{-1}(\bar{\boldsymbol{x}}')$. In this case, given either term in $L$ being 0, the other term will be positive due to the identity property of $\ell(\cdot)$. Correspondingly, $L = \ell(f_\theta(\tilde{\mathcal{A}}), \bar{\boldsymbol{x}}) + \ell(f_\theta(\tilde{\mathcal{A}}), (\pi^v)^{-1}(\bar{\boldsymbol{x}}')) > 0$.

## A.7 Additional numerical results

we expand our analysis to incorporate two additional evaluation metrics beyond Top-$m\%$ errors as follows.

**Objective values ($\downarrow$) comparing to the ILP solver** Instead of directly using the rounded solutions for calculations—since they are often infeasible—we leverage these predictions as initial points to expedite the solving process of ILP solvers(Nair et al., 2020; Khalil et al., 2022; Han et al., 2023). Consistent with established practices, we integrate predictions from the trained models to enhance

the ILP solver CPLEX, following the methodology outlined in (Nair et al., 2020). The results are shown in Table 3 and Table 4 for datasets BIP and SMSP, respectively. Note that the dataset BPP is excluded, as its instances can be solved in a matter of seconds.

Table 3: Objective values v.s. solving time on BIP

|  | 100s | 200s | 300s | 400s | 500s | 600s |
|---|---|---|---|---|---|---|
| CPLEX | 16.9 | 16.2 | 15.8 | 15.5 | 15.3 | 15.2 |
| CPLEX + "Orbit+" | **15.5** | **15.1** | **14.9** | **14.8** | **14.7** | **14.6** |

Table 4: Objective values v.s. solving time on SMSP

|  | 100s | 200s | 300s | 400s | 500s | 600s |
|---|---|---|---|---|---|---|
| CPLEX | 37.8 | 25.2 | 13.6 | 12.1 | 11.3 | 11.0 |
| CPLEX + "Orbit+" | **13.7** | **13.1** | **12.3** | **11.6** | **11.1** | **10.7** |

From these results, we observe that our augmentation scheme yields better objective values while requiring less computational time.

**Constraint violations** ($\downarrow$)   We also report the total violation of model prediction $\hat{x}$ with respect to the constraint $A\hat{x} \leq b$, i.e., the summation of positive elements in $A\hat{x} - b$. The violation of predictions from different models is summarized in Table 5.

Table 5: Constrain violations.

|  | BIP | SMSP |
|---|---|---|
| Uniform | 27.84 | 852.23 |
| Position | 22.12 | 933.76 |
| Orbit | 12.89 | 453.87 |
| Orbit+ | **3.68** | **329.74** |

From the results, we find that our methods (Orbit, Orbit+) produce predictions with significantly less constraint violation, further demonstrating the effectiveness of our approach.

### A.8 SYMMETRY DETECTION

Detecting a symmetry group $\mathcal{G}$ for an ILP is complex and can be computationally intensive. Fortunately, over the years, well-established methods and software tools such as Bliss(Junttila & Kaski, 2011) and Nauty(McKay & Piperno, 2013) have been developed for efficiently detecting the symmetries of ILPs, as well as their orbits. In our experiments, the orbits of all instances have been detected. In Table 6, we report the average time taken to detect symmetry groups in the considered datasets using Bliss, with the size of the detected subgroup $G$ represented by its logarithmic value $\log_{10} |G|$.

Table 6: Average time of symmetry detection.

|  | BPP | BIP | SMSP |
|---|---|---|---|
| # of Var. | 420 | 1083 | 23000 |
| $\log_{10} |G|$ | 6.9 | 6.6 | 213.1 |
| time(s) | 0.05 | 0.06 | 6.11 |

From the results, we observe that for smaller problems like BPP and BIP, symmetry detection requires negligible computational time. Even for larger problems, such as SMSP, symmetry group detection is still accomplished within a few seconds, demonstrating the feasibility of our approach even for complex instances.

