# OpenReview forum: "When GNNs meet symmetry in ILPs: an orbit-based feature augmentation approach"
_ICLR.cc/2025/Conference — ICLR 2025 Poster_

### Official Review · Reviewer_EWnQ · 2024-11-03

**Soundness:** 3
**Presentation:** 2
**Contribution:** 3
**Rating:** 8
**Confidence:** 2

**Summary:**

The authors propose a novel feature augmentation method for ILP's with symmetries that are described by bipartite graphs for solving with GNN's. The augmentations obey some important symmetry properties but are also more parsimonious than existing methods. Empirical results suggest that these augmentations help the GNN's break the symmetry better than competing methods.

**Strengths:**

The main idea in Section 4.2 is explained well. The numerical results are decent and convey the practical benefits of the method. The introduction is also nicely written.

**Weaknesses:**

While the numerical results do suggest that the method helps break symmetries better than others, it is hard to tell if this makes a difference in the end result (objective of rounded solution). I think such a comparison should be added (either way). A more complete description or explanation of the augmentation procedure in general could be useful for those not in the field. Some minor improvement for minor writing weaknesses are suggested below.

**Questions:**

Figure 1 appears too soon, before the map between ILP and graphs...
I don't think you need to define a permutation.
I don't understand the h_i^v definitions in 2.3, and how they're related to c, b, and A in the original problem? Isn't c_i a scalar? How is this a feature vector? Can you please clarify the entries of \mathcal{A} and how they relate to the bipartite graph? (Are you using the columns and rows of \mathcal{A} as the feature vectors for the nodes V and W? If this is true, I think it should be explicitly stated somewhere.)
I think you mean {V,W,E} as the bipartite graph? (You have C in one place and W in another)
In figure 1, I don't yet understand why the GNN must make x1 and x2 equal? How was the integer constraint enforced?
The notation in proposition 1 seems a bit overkill - this seems like a simple & intuitive result, so perhaps this proof can be made simpler.
Am I missing an additional assumption of Proposition 1 that the initial feature embedding of points in the same orbit are equal? (Otherwise this proposition extends to the proposal in section 4?)
Corollary 1 seems overly general (and not quite correct). I think you mean that it cannot ALWAYS predict the optimal solution. Of course there are instances that it can solve correctly (like min 0 s.t. [no constraints])
The number of instances in the problems 5.1 is quite low, isn't it?
How is the GNN enforcing integer constraints?
Do you have any other metrics of comparison? Don't you also care if the rounded solution is any good? (It's OK if it's not -- I'm just curious.)
Minor: "By some mathematical derivations" is an awkward phrase; typo "cloest" on page 9.

---

> ### Author Response · Authors · 2024-11-26
> **Rebuttal Part I**
>
> >### **Weakneas**
> >#### **W1. While the numerical results do suggest that the method helps break symmetries better than others, it is hard to tell if this makes a difference in the end result (objective of rounded solution). I think such a comparison should be added (either way).**
>
> Thank you for your constructive suggestion. In response, we have expanded our analysis to incorporate two additional evaluation metrics beyond $\ell_1$ distances:
>
>
> 1. **Objective values ($\downarrow$)**: **Instead of directly using the rounded solutions for calculations—since they are often infeasible—we leverage these predictions as initial points to expedite the solving process of ILP solvers [1,2,3]**. Consistent with established practices, we integrate predictions from the trained models to enhance the ILP solver CPLEX, following the methodology outlined in [1]. Note that the dataset BPP is excluded, as its instances can be solved in a matter of seconds.
>
>     - **Dataset BIP**
>
>     |  | 100s | 200s | 300s | 400s | 500s | 600s |
>     |-|:-:|:-:|:-:|:-:|:-:|:-:|
>     | Uniform  | 16.8 | 16.1 | 15.9 | 15.5 | 15.4 | 15.3 |
>     | Position  | 16.2 | 15.8 | 15.5 | 15.2 | 15.0 | 14.9 |
>     | Orbit+ | **15.5** | **15.1** | **14.9** | **14.8** | **14.7** | **14.6** |
>
>     - **Dataset SMSP**
>
>     | | 100s | 200s | 300s | 400s | 500s | 600s |
>     |-|:-:|:-:|:-:|:-:|:-:|:-:|
>     | Uniform| 37.7 | 25.1 | 13.5 | 12.2 | 11.2 | 11.0 |
>     | Position  | 22.7 | 15.2 | 13.0 | 11.7 | 11.3 | 10.9 |
>     | Orbit+| **13.7** | **13.1** | **12.3** | **11.6** | **11.1** | **10.7** |
>
>     From these results, we observe that **our augmentation scheme yields better objective values while requiring less computational time**.
>
> 2. **Constraint violations ($\downarrow$)**: We also report the total violation of model prediction $\hat{x}$ with respect to the constraint $A\hat{x} \leq b$, i.e., the summation of positive elements in $A\hat{x} - b$. The violation of predictions from different models is summarized below.
>
>     | |BIP|SMSP|
>     |:-:|:-:|:-:|
>     |Uniform|27.84|852.23|
>     |Position|22.12|933.76|
>     |Orbit|12.89|453.87|
>     |Orbit+|**3.68**|**329.74**|
>
>     From the results, we find that **our methods (Orbit, Orbit+) produce predictions with significantly less constraint violation**, further demonstrating the effectiveness of our approach.
>
> We hope these additional metrics provide a more comprehensive evaluation of the performance of our method and address your suggestion.
>
>
> >#### **W2. A more complete description or explanation of the augmentation procedure in general could be useful for those not in the field. Some minor improvement for minor writing weaknesses are suggested below.**
>
> Thank you for your valuable feedback. In response to this suggestion, along with the related points raised in the question section, we are revising our manuscript to incorporate these enhancements and will upload the updated version accordingly.
>
>
> >### **Qestions**
>
> We apologize for any confusion caused by our notations and descriptions. Below, we provide clarifications to address these questions.
>
> >#### **Q1. Figure 1 appears too soon, before the map between ILP and graphs.**
>
> Thank you for your suggestion. We have moved Figure 1 to Section 3, after the introduction of ILP and bipartite graph concepts.
>
>
>
> >#### **Q2.  I don't understand the h_i^v definitions in 2.3, and how they're related to c, b, and A in the original problem? Isn't c_i a scalar? How is this a feature vector?**
>
>
> Thank you for your question. $c_i$ and $h_i^v$ are both scalars. The term "feature vector" in this context is misleading, and we apologize for the confusion.
>
> Here is the corrected explanation of the relations between the bipartite graph and the problem coefficients $A\in \mathbb{R}^{m\times n}, b\in\mathbb{R}^m$ and $c\in\mathbb{R}^n$.
>
>
> The bipartite graph representation $\\{V,W,E\\}$ consists of:
> - Variable nodes $V:=\\{v_1,\dots,v_n\\}$, where each variable node $v_j$ has a feature $h^v_j:=c_j\in\mathbb{R}$. This corresponds to the scalar coefficients from the objective function.
> - Constrain nodes $W:=\\{w_1,\dots,w_m\\}$, where each constraint node $w_i$ has a feature $h^w_i:=b_i\in\mathbb{R}$. This corresponds to the right-hand side values of the constraints.
> - Edges $E:=\\{e_{ij}|A_{ij}\neq 0\\}$, where $e_{ij}$ represents the edge between constraint node $i$ and variable node $j$, and each edge has a feature $h^e_{ij}:=A_{ij}\in \mathbb{R}$. This corresponds to the matrix coefficients in the constraints.
>
> In short, $A,b,c$ are used as edge features, constraint features, and variable features, respectively.

---

> ### Author Response · Authors · 2024-11-26
> **Rebuttal Part II**
>
> >#### **Q3. Can you please clarify the entries of \\mathcal{A} and how they relate to the bipartite graph? (Are you using the columns and rows of \\mathcal{A} as the feature vectors for the nodes V and W? If this is true, I think it should be explicitly stated somewhere.)**
>
> The entries of $\\mathcal{A}$ represent different feature types associated with the nodes and edges of a bipartite graph. Specifically, each row (or column) in $\\mathcal{A}$ includes both edge features and constraint (or variable) features. We elaborate on this below.
>
> A bipartite graph consists of two sets of nodes: $V=\\{v\_1,\\dots,v\_n\\}$ (variable nodes) and $W=\\{w\_1,\\dots,w\_m\\}$ (constraint nodes), connected by edges. Each node and each edge in this graph is associated with a feature. We can put all features together in the following form:
> $\\begin{array}{c|cccc}
>      & v\_1 & \\cdots & v\_n &  \\\\
> \\hline
> w\_1 & h^e\_{11} & \\cdots & h^e\_{1n} & h^w\_{1} \\\\
> \\cdots & \\cdots & \\cdots & \\cdots & \\cdots \\\\
> w\_m & h^e\_{m1} & \\cdots & h^e\_{mn} & h^w\_{m} \\\\
>  & h^v\_{1} & \\cdots & h^v\_{n} & 0 \\\\
> \\end{array}$
>
> In this table:
>
> - Each constraint node $w\_i$ corresponds to one row, i.e., $w\_i \\rightarrow \\begin{bmatrix}h^e\_{i1}&\\cdots& h^e\_{in}&h^w\_i\\end{bmatrix}$.
> - Each variable node $v\_j$ corresponds to one column, i.e., $v\_j \\rightarrow\\begin{bmatrix}h^e\_{1j} &\\cdots& h^e\_{mj}&h^v\_j\\end{bmatrix}^\\top$.
>
> We can thus use the feature matrix $\\mathcal{A}$ to represent the bipartite graph.
> According to responses to Q2, $h^e\_{ij}:=A\_{ij}\\in\\mathbb{R}$, $h^w\_i:=b\_i\\in\\mathbb{R}$ and $h\_j^v:=c\_j\\in\\mathbb{R}$, then we have
>
> $\\begin{bmatrix}h^e\_{11} & \\cdots & h^e\_{1n}&h^w\_{1}\\\\
> \\cdots&\\cdots & \\cdots &\\cdots \\\\
> h^e\_{m1} & \\cdots & h^e\_{mn}&h^w\_{m}\\\\
> h^v\_1 & \\cdots &h^v\_n & 0  \\end{bmatrix}=\\begin{bmatrix}A\_{11} & \\cdots & A\_{1n}&b\_1\\\\
> \\cdots&\\cdots & \\cdots &\\cdots \\\\
> A\_{m1} & \\cdots & A\_{mn}&b\_m\\\\
> c\_1 & \\cdots &c\_n & 0  \\end{bmatrix}=\\begin{bmatrix}A&b\\\\c^\\top & 0 \\end{bmatrix}=\\mathcal{A}\\in\\mathbb{R}^{(m+1)\\times (n+1)}$
>
> Thus, $\\mathcal{A}\\in \\mathbb{R}^{(m+1)\\times (n+1)}$ contains three types of features:
> - Edge features in the top-left submatrix of dimension $m\\times n$,
> - Variable features in the last row (excluding the last entry),
> - Constraint features in the last column (excluding the last entry).
>
>
> >#### **Q4.  I think you mean {V,W,E} as the bipartite graph? (You have C in one place and W in another)**
> Yes, you are correct. The letter $C$ is a typo. We use $W$ to represent the set of constraint nodes in the bipartite graph, not $C$. Thank you for pointing out that.
>
> >#### **Q5.  In figure 1, I don't yet understand why the GNN must make x1 and x2 equal?**
>
> That's because the $x\_1$ and $x\_2$ are symmetric. More specifically, in the bipartite graph, the nodes corresponding to $x\_1$ and $x\_2$ are automorphic. Conceptually, from the GNN's perspective, **these nodes share the same features and neighborhoods**, so their outputs must be identical.
>
> >#### **Q6. How was the integer constraint enforced?**
>
> The integer constraint cannot be enforced by GNN itself. Instead, **it is approximately enforced by minimizing the loss function**, which includes a target "integer" label.
>
> >#### **Q7. The notation in proposition 1 seems a bit overkill - this seems like a simple & intuitive result, so perhaps this proof can be made simpler.**
>
> Thank you for your feedback. While the notation may appear verbose, we believe it is the simplest and most precise way to express the result clearly and rigorously.
>
>
> >#### **Q8. Am I missing an additional assumption of Proposition 1 that the initial feature embedding of points in the same orbit are equal? (Otherwise this proposition extends to the proposal in section 4?)**
>
> No, there is no assumption beyond Assumption 1 and the formulation symmetry in the ILP. **Implicitly**, if the ILP exhibits formulation symmetries, then the variables within the same orbits, with respect to these symmetries, must have the same initial features.
>
> >#### **Q9. Corollary 1 seems overly general (and not quite correct). I think you mean that it cannot ALWAYS predict the optimal solution. Of course there are instances that it can solve correctly (like min 0 s.t. [no constraints])**
>
> You are absolutely right. Thank you for pointing out this issue. We have revised the corollary in the updated manuscript.
>
> >#### **Q10. The number of instances in the problems 5.1 is quite low, isn't it?**
>
> Yes, the datasets used are relatively small. However, the evaluation results are promising. We believe this is due to the GNN architecture's strong generalization capabilities, which enable it to perform well even on small datasets.

---

> > ### Author Response · Authors · 2024-11-26
> > **Rebuttal Part III**
> >
> > >#### **Q11. How is the GNN enforcing integer constraints?**
> >
> > For a detailed explanation, please refer to our response to Q6.
> >
> > >#### **Q12. Do you have any other metrics of comparison?**
> > Yes, we have added two additional metrics. Please refer to our response to W1 for more details.
> >
> > >#### **Q13. Don't you also care if the rounded solution is any good? (It's OK if it's not -- I'm just curious.)**
> >
> > You raise an important point. The rounded solution is often infeasible, and existing literature typically applies post-processing techniques to the predicted solution to identify feasible ones. We have added such experiments in our response to W1, please refer to it.
> >
> > ---
> > ### **References**
> >
> > [1] Nair V, Bartunov S, Gimeno F, et al. Solving mixed integer programs using neural networks[J]. arXiv preprint arXiv:2012.13349, 2020.
> >
> > [2] Khalil E B, Morris C, Lodi A. Mip-gnn: A data-driven framework for guiding combinatorial solvers[C]//Proceedings of the AAAI Conference on Artificial Intelligence. 2022, 36(9): 10219-10227.
> >
> > [3] Han Q, Yang L, Chen Q, et al. A gnn-guided predict-and-search framework for mixed-integer linear programming[J]. arXiv preprint arXiv:2302.05636, 2023.

---

> > > ### Comment · Reviewer_EWnQ · 2024-11-27
> > > **Thank you for responses**
> > >
> > > Thank you for the additional experiments and clarifications. I have increased my score.

---

> > > > ### Author Response · Authors · 2024-11-30
> > > >
> > > > Thank you very much for your thoughtful feedback and for increasing your score. I truly appreciate the time and effort you dedicated to reviewing my work, as well as your constructive comments. Your support means a great deal, and it has been a pleasure to revise the manuscript based on your insights.

---

### Official Review · Reviewer_feSQ · 2024-11-04

**Soundness:** 2
**Presentation:** 3
**Contribution:** 2
**Rating:** 5
**Confidence:** 4

**Summary:**

This paper proposes to augment the vertice features in ILP graph representation based on orbit of the symmetry group. Some theory is provided and numerical results are conducted with the comparison with the random feature technique (Chen et al. 2022) and the positional ID technique (Han et al. 2023).

**Strengths:**

1. The paper is in general well-written and is easy to follow.

2. The idea of augmenting features based on orbits is reasonable. I agree with the authors that only vertices in the same orbit need to be separated with augmented features, and the cardinality of the augmented feature space is much smaller than $n!$ if the number of orbits is much larger than one.

3. The reported numerical results look better than baseline methods.

**Weaknesses:**

1. There is no comparison with conventional methods based on symmetry group and orbits, such as Ostrowski et al. (2011) cited by the authors. In addition to Ostrowski et al. (2011), there should actually be a much richer literature in this direction.

2. There is no report on the cost of computing the symmetry group. I expect to see a trade-off between the size of the symmetry group and the improvement from "no augmentation".

**Questions:**

None

---

> ### Author Response · Authors · 2024-11-25
> **Rebuttal Part I**
>
> >#### **W1.1 There is no comparison with conventional methods based on symmetry groups and orbits, such as Ostrowski et al. (2011) cited by the authors.**
>
> Thank you for raising this point. Indeed, a comparison with conventional symmetry-breaking methods is essential. To address this, we have conducted additional experiments to benchmark our approach against these methods. Since classic symmetry-breaking techniques, such as those outlined by Ostrowski et al. (2011), are already implemented in CPLEX, our comparison focuses on the following aspects:
>
> - **conventional**: ILP solver CPLEX with its hyperparameters tuned as "highest level of symmetry breaking", where multiple conventional symmetry breaking algorithms such as orbitope fixing, isomorphic prunning and orbital branching are applied adaptively.
> - **ours**: Default CPLEX without emphasizing symmetry breaking, while warm-started by the predictions from "Orbit+" approach.
>
> We applied these two methods to solve instances of datasets BIP and SMSP with a time limit of 600s. The **objective values ($\\downarrow$)** at different times are reported below.
>
> - **For dataset BIP**
>
>     |               | 100s | 200s | 300s | 400s | 500s | 600s |
>     |----------------------------|:----:|:----:|:----:|:----:|:----:|:----:|
>     | conventional                       | 16.9 | 16.2 | 15.8 | 15.5 | 15.3 | 15.2 |
>     | ours                       | **15.5** | **15.1** | **14.9** | **14.8** | **14.7** | **14.6** |
>
> - **For dataset SMSP**
>
>     |               | 100s | 200s | 300s | 400s | 500s | 600s |
>     |----------------------------|:----:|:----:|:----:|:----:|:----:|:----:|
>     | conventional                      | 37.8 | 25.2 | 13.6 | 12.1 | 11.3 | 11.0 |
>     | ours                       | **13.7** | **13.1** | **12.3** | **11.6** | **11.1** | **10.7** |
>
>
> According to the results, **our method shows better objective values** across all solving time compared to the "conventional" baseline.
>
> >#### **W1.2. In addition to Ostrowski et al. (2011), there should actually be a much richer literature in this direction.**
>
> Thanks for your suggestion, we have added a more comprehensive literature review in the revised manuscript. The updated paragraph is as follows:
>
> To address this, several approaches have been developed. Margot (2002) prunes the enumeration tree in the branch and bound algorithm; Ostrowski et al. (2008; 2011) propose branching strategies based on orbits; Puget (2003; 2006) enhances problem formulations by introducing symmetry-breaking constraints. A more comprehensive survey of related research is provided in (Margot, 2009).

---

> ### Author Response · Authors · 2024-11-25
> **Rebuttal Part II**
>
> >#### **W2.1 There is no report on the cost of computing the symmetry group.**
>
> Thank you for the valuable comment. In our experiments, the symmetry groups are detected by the well-established tool, Bliss [1]. It integrates efficient group detection algorithms that can heuristically identify symmetry subgroups within a relatively small amount of time. To illustrate this, we supplement the average time taken to detect symmetry groups in the considered datasets using Bliss, with the size of the detected subgroup $G$ represented by its logarithmic value $\\log\_{10} |G|$
>
>
> |       |  BPP |  BIP |  SMSP |
> |:------------:|:----:|:----:|:-----:|
> | # of Var. |  420 | 1083  | 23000 |
> | $\\log\_{10} \|G\|$ |  6.9 |  6.6 | 213.1 |
> |    time(s)   | 0.05 | 0.06 |  6.11 |
>
> From the results, we observe that for smaller problems like BPP and BIP, symmetry detection requires negligible computational time. Even for larger problems, such as SMSP, symmetry group detection is still accomplished within a few seconds, demonstrating the feasibility of our approach even for complex instances.
>
> >#### **W2.2 I expect to see a trade-off between the size of the symmetry group and the improvement from "no augmentation".**
>
> Thanks for your suggestion.  We include additional results showing the top-m% error ($\\downarrow$) for models trained with different sizes of symmetry groups.
>
> Here, **No-Aug** denotes "no augmentation," and **sym-k%** refers to a subgroup with k% of the generators of the detected symmetry group. **As k increases, the size of the symmetry group used in training increases.**
>
> The table below shows the results for datasets BPP, BIP, and SMSP:
>
> |           |       |  BPP  |       |   |        |   BIP  |        |   |        |  SMSP  |         |
> |:---------:|:-----:|:-----:|:-----:|---|:------:|:------:|:------:|---|:------:|:------:|:-------:|
> | Top error |  30%  |  50%  |  90%  |   |   30%  |   50%  |   90%  |   |   30%  |   50%  |   90%   |
> |   No-Aug  | 3.00  | 4.80  | 9.50  |   | 30.40  | 50.60  | 91.10  |   | 34.00  | 57.70  | 113.90  |
> |  sym-60%  | 1.07  | 2.39  | 9.37  |   | 17.09  | 32.04  | 86.33  |   | 15.57  | 30.91  |  90.69  |
> |  sym-70%  | 0.73  | 1.75  | 8.69  |   | 14.57  | 26.54  | 85.08  |   | 11.90  | 24.54  |  81.69  |
> |  sym-80%  | 0.35  | 1.36  | 7.29  |   | 11.08  | 20.79  | 83.84  |   |  8.36  | 18.58  |  74.14  |
> |  sym-90%  | 0.22  | 0.92  | 5.81  |   |  6.48  | 13.69  | 81.47  |   |  4.67  | 10.89  |  63.70  |
> |  sym-100% | 0.00  | 0.00  | 4.30  |   |  3.20  |  5.50  | 79.20  |   |  0.00  |  1.00  |  50.30  |
>
> From the results, it is clear that **increasing the size of the symmetry group leads to greater improvement across all datasets**. While, as shown in W2.1, a larger size of the symmetry group generally means a higher detection cost, **the trade-off lies in the balance between the computational cost of detecting larger symmetry groups and the performance gain achieved through the use of more comprehensive symmetry information in training.**
>
> ---
> ### **References**
>
> [1] Junttila T, Kaski P. Conflict propagation and component recursion for canonical labeling[C]//International Conference on Theory and Practice of Algorithms in (Computer) Systems. Berlin, Heidelberg: Springer Berlin Heidelberg, 2011: 151-162.

---

> > ### Author Response · Authors · 2024-12-03
> > **Kind reminder**
> >
> > Dear Reviewer feSQ,
> >
> > Thank you for your valuable suggestions. Our work has greatly benefited from your feedback. **As the discussion phase is coming to a close, we would like to ask if you have any further comments or suggestions.** We are keen to make any last improvements to our work.
> >
> > Thank you again for your time and consideration.
> >
> > Best regards,
> >
> > The Authors

---

### Official Review · Reviewer_GLPM · 2024-11-04

**Soundness:** 3
**Presentation:** 3
**Contribution:** 2
**Rating:** 6
**Confidence:** 5

**Summary:**

This paper improves a weakness of graph network approaches for predicting the solutions of integer linear programs (ILPs) with symmetric variables. Because graph networks are permutation equivariant, they cannot distinguish between exchangeable variables in the ILPs (or more concretely, variables such that, when permuted, the cost constraint is still satisfied and the objective is unchanged). The authors use feature augmentation to break the symmetries between these equivalent variables, specifically emphasizing distinguishability, “isomorphic consistency”, and “augmentation parsimony”. Past works have used feature augmentation to break these symmetries, but without adhering to these principles. They demonstrate the effectiveness of their techniques relative to alternatives on solving synthetic ILPs, with training data generated by a classical solver.

**Strengths:**

This paper addresses a meaningful issue (symmetry-breaking in integer linear programs) in a new way (using inputs to GNNs that break only the required orbit symmetries, in accordance with the three defined desiderata). Although I believe these two pieces are individually not new (see “weaknesses”), their combination is. The “Orbit+” approach is also a novel way of enhancing “augmentation parsimony”. The paper is generally clear, and the writing style and notations are both enjoyable to read. Their experimental results on the chosen tasks beat the chosen ML baselines. Although I am not convinced by the necessity of isomorphic consistency for most applications, in which a logical loss function can be chosen (which is unaffected by swapping equivalent nodes), the use of SymILO to enforce it by adjusting training labels is also new.

**Weaknesses:**

I think the biggest weakness of this paper is that, in short, many of its central ideas have already been introduced and (in some cases) thoroughly explored in papers of which the authors seem unaware. (This is understandable, given that they do not appear in the ILP literature and use different terminology to describe the problem, but nonetheless they exist — I hope the authors may be inspired by the perspectives of these papers, and can also articulate the novelty of their work relative to them.) This line of work (see the references below; although not the earliest, [2] or [4] may be the most accessible starting points) goes by the name “symmetry-breaking”, and articulates the precise issue that the authors encounter for ILPs, but in a much more general way, for all group equivariant networks. The principles of distinguishability and augmentation parsimony are explored under different names, e.g. in [3]. There is a related line of work on breaking symmetries of sets, termed “multiset equivariance” [5]. Works such as these and [4] make clear subtleties of the problem that aren’t discussed in this paper, such as the difference between the graph automorphism group and the node orbits, and articulate methods for addressing the equivalent nodes of ILPs in ways that subsume the method presented here.

I believe that orbit-equivariant graph neural networks [6] are also a slightly more fleshed out version of the “Orbit” approach.

As noted under questions, I also find the discussion of isomorphic consistency confusing, as it is (assuming I understand correctly) not well-motivated under orbit-invariant loss functions, and importantly, not necessarily even possible to achieve. Is “relaxed equivariance” [2] more suitable?

Finally, as also noted under questions, there seem to be weaknesses with the experiments — namely, the choice of loss function, and the premise/lack of comparison to non-ML baselines.

References:
1. Smidt, T. E., Geiger, M., and Miller, B. K. Finding symmetry breaking order parameters with euclidean neural networks. Phys. Rev. Research, 3: L012002, Jan 2021. doi: 10.1103/PhysRevResearch
2. Kaba, S.-O. and Ravanbakhsh, S. Symmetry breaking and equivariant neural networks. In Symmetry and Geometry in Neural Representations Workshop, NeurIPS, 2023.
3. Xie, Y. and Smidt, T. Equivariant symmetry breaking sets. TMLR 2024.
4. Hannah Lawrence, Vasco Portilheiro, Yan Zhang, and Sekou-Oumar Kaba. Improving equivariant networks with probabilistic symmetry breaking. In ICML 2024 Workshop on Geometry-grounded Representation Learning and Generative Modeling, 2024.
5. Zhang, Y., Zhang, D. W., Lacoste-Julien, S., Burghouts, G. J., and Snoek, C. G. M. Multiset-equivariant set prediction with approximate implicit differentiation. In International Conference on Learning Representations, 2022.
6. Morris, M., Grau, B. C., & Horrocks, I. (2024). Orbit-equivariant graph neural networks. ICLR 2024.

**Questions:**

Questions:
1. *Value of ML versus classical ILP method*: The authors motivate the use of GNNs for ILPs by referencing recent works which use GNNs as part of the solution process, e.g. as an oracle in branch-and-bound algorithms or predicting initial solutions. However, the experiments they run directly output solutions of the input ILPs, and there is no comparison to classical ILP solvers (eg int terms of accuracy), because the training data itself is generated by the ILP solver SCIP. Thus, my question is: what value does machine learning add to these problems? For example, are the trained models’ forward passes on test data supposed to be faster than the solver? (If so, does this take into account the time to detect the symmetry of the input ILP?) Does the noted symmetry-breaking problem arise when ML is used to predict branching decisions or node selections?
2. *Choice of loss function in experiments*: why use a loss function that tries to learn the exact solution instance, when there are several degenerate, “equally good” solutions? It seems far more natural to use the objective function $c^Tx$ of the ILP itself, or the fraction of instances that satisfy the constraint $Ax\leq b$, over $x$ from the test set.
3. *Motivation for “isomorphic consistency” principle*: The isomorphic consistency principle (which is dataset-dependent) strikes me as odd — the whole problem explored in this paper is that it is not even **possible** to satisfy while outputting optimal solutions, except for on certain training sets. Can the authors elaborate on this? IN particular, if one uses a loss function like the one suggested above, I can’t tell what the value of isomorphic consistency is.
4. *Practicalities of the experiments*: Line 468 states that multiple augmentations need to be drawn per sample. Is this done for baselines too? Also, is there an ablation result for the importance of using SymILO to enforce isomorphic consistency? Does this remain under the choice of loss function suggested above?
5. *Minor clarification about formulation symmetry definition*: As defined around line 118, a formulation symmetry “retains the description $Ax \leq b$”. One way of achieving this is if $Ag(x)=Ax$, indeed this is what I would have expected as the definition. Is this more accurate?
In the methodology paragraph, around lines 228-230, the paper says “…the approach did not exploit the underlying symmetry properties, leading to suboptimal performance on ILPs with strong symmetries”. What does this mean? The paper would be improved by making this claim precise, and including references to evidence (eg a specific result in the cited paper).
6. Why not use MIPLIB, the dataset cited in the first paragraph of the paper, in the experiments? For the datasets used in experiments, what fraction of inputs exhibit no symmetric variables at all?

Typos:
As I read through the paper, I noticed some minor typos. These did not affect my evaluation of the paper, but I’m noting them here in case it’s useful to the authors.
* line 74: “fully use of” —> “full use of”
* Subtitle on line 191: “issues occur”, not “occurred”
* Line 207: “correspond”, not “corresponds”
* Line 218: “oribit”
* Line 219: “have distinct values”, not “has distinct values”
* Line 445: “cloest”

Minor notation/writing notes:
* Definition 3 should define that $I \in I^n$ and should also define $\mathcal{O}_i$.
* The standard term for Assumption 1 is permutation equivariance, not equivalence
* It would be good to explain lines 322-323 (“Accordingly…other orbits”) in mathematically precise language

On the writing side, I would also recommend making the “Motivations” paragraph more concrete.

---

> ### Author Response · Authors · 2024-11-25
> **Rebuttal Part I**
>
> >### **W1. Concerns about novelty**
> >#### **W1.1. ... many of its central ideas have already been introduced and (in some cases) thoroughly explored in papers of which the authors seem unaware... I hope the authors ... can also articulate the novelty of their work relative to them.**
>
> Thank you very much for providing the detailed references [1-6], which investigate problems related to the "indistinguishability" issue we address in ILPs (i.e., GNNs' inability to distinguish symmetric variables). We have added a paragraph in the revised manuscript to acknowledge and cite these relevant works. Below, we summarize the novelty of our work in comparison to theirs.
>
> 1. **Novelty in the Context of ILPs**: To our best knowledge, our work is the first to identify and address the "indistinguishability" issue specifically in the context of ILPs.
> 2. **Our work utilizes special problem structures in ILPs**.
>     - Our approach **introduces augmented features only to the variable nodes** in the bipartite graph representation of ILPs. This targets the variable symmetries, as they are sufficient for differentiating outputs during solution prediction, and disregards the constraint symmetries. Compared to existing methods, this strategy simplifies the symmetry groups that need to be processed, making the approach more efficient.
>     - We also propose **reducing the space of augmented features by leveraging structures formed by different orbits**, as detailed in our "Orbit+" approach (Section 4.2 of the manuscript). In contrast, prior methods either do not make use of orbits or fail to adequately consider the relationships between different orbits, which limits their effectiveness in distinguishing symmetries.
> 3. In addition to addressing the "indistinguishability" issue, **our work also enhances model training by enforcing "isomorphic consistency"**, which has not been explored in those referenced works.
>
>
>
> >#### **W1.2. I believe that orbit-equivariant graph neural networks [6] are also a slightly more fleshed out version of the “Orbit” approach.**
>
> Thanks for pointing out this reference. Paper [6] proposes two methods based on Orbit: Orbit-Inv-GNN and m-Orbit-Transform-GNN. While m-Orbit-Transform-GNN is presented as an extension of Orbit-Inv-GNN, its performance is consistently and significantly worse than that of Orbit-Inv-GNN in their main results. As a result, we compare our method to Orbit-Inv-GNN and highlight the following relations and differences:
>
> 1. For the generation of "symmetry breaking" features, **our "Orbit" method = Orbit-Inv-GNN + Randomlization**.
>
> In the case of an orbit of size $m$, **Orbit-Inv-GNN deterministically assigns IDs $\\{1,\\dots,m\\}$** to the corresponding nodes. In contrast, **our Orbit method randomly samples a shuffled ordering.** This randomization prevents the model from overfitting to deterministic values and improves generalization performance. To illustrate this, we conducted a comparative experiment on the BP dataset. The training and validation losses ($\\downarrow$) for the deterministic approach ("Det"), "2x sampling", and "8x sampling" are as follows:
>
> |  epoch |   10  |   30  |   50  |   70  |   90  |
> |:------|:-----:|:-----:|:-----:|:-----:|:-----:|
> |  Det-train | 0.24  | 0.22  | 0.21  | 0.20  | 0.18  |
> | Det-valid | 0.24  | 0.23  | 0.24  | 0.24  | 0.25  |
> |  2x-train | 0.24  | 0.23  | 0.22  | 0.22  | 0.21  |
> | 2x-valid | 0.25  | 0.23  | 0.23  | 0.23  | 0.23  |
> |  8x-train | 0.24  | 0.23  | 0.22  | 0.21  | 0.21  |
> | 8x-valid | 0.24  | 0.23  | 0.22  | 0.22  | 0.22  |
>
> From the results, we can clearly observe that the **deterministic approach suffers severer overfitting than the stochastic ones** with the epoch number increasing.
>
> 2. Our "Orbit+" approach is a more fleshed-out version of Orbit-Inv-GNN
>
> Except for randomization, another significant difference is that our enhanced version "Orbit+" leverages special structures among orbits (please refer to Section 4.2 of the manuscript) to improve the performance, while Orbit-Inv-GNN doesn't.
>
> Moreover, such structures are commonly frequently observed in ILPs. For example, paper [7] analyzes the symmetries of the real-world MIBLIB benchmark and finds that more than half of the symmetries exhibit such structures (referred to as "matrix action"). This highlights the practical value of our "Orbit+" method

---

> ### Author Response · Authors · 2024-11-25
> **Rebuttal Part II**
>
> >### **W2. Confusion about the principle of isomorphic consistency**
> >#### **W2.1. As noted under questions, I also find the discussion of isomorphic consistency confusing, as it is (assuming I understand correctly) not well-motivated under orbit-invariant loss functions**
>
> Thanks for pointing out this and we should have illustrated it more clearly. Below, we provide additional clarifications.
>
> 1. **Isomorphic consistency is specifically designed for supervised learning tasks**, as mentioned earlier in the preliminaries section of the manuscript. "Orbit-invariant" loss functions fall outside the scope of this discussion.
> 2. To the best of our knowledge, there is **no research that applies the orbit-invariant loss (assume it is constructed through objective functions and constraint violations) for general ILPs. Supervised learning remains a critical component of current studies** [11-14]. In existing research, "orbit-invariant" loss functions are typically employed for continuous problems [8] or specialized classes of discrete problems like QUBO [9] and TSP [10].
> 3. A potential reason for the lack of application of "orbit-invariant" losses to general ILPs is **the challenge of enforcing integrality constraints**. If the integrality constraints are ignored, the model effectively reduces to solving LPs, rather than ILPs. In contrast, supervised learning allows for the approximate enforcement of integer constraints by minimizing prediction errors with respect to labeled data.
>
>
>
> >#### **W2.2. ... and importantly, not necessarily even possible to achieve. Is “relaxed equivariance” [2] more suitable?**
>
> - **Theoretically it is achivable.** Assume $\\bar{x}$ is an feasible solution of $\\mathcal{A}$. Since $\\exists \\pi,\\sigma$ such that $\\pi^c(\\sigma^r(\\mathcal{A}))=\\mathcal{A}'$, its easy to verify that $\\pi^v(\\bar{x})$ is a feasible solution of $\\mathcal{A}'$. Let $\\bar{x}$ be the label of $\\mathcal{A}$ and  $\\pi^v(\\bar{x})$ be that of $\\mathcal{A}'$, then the sufficient condition in this principle holds.
> - **Practically, it can be approximately enforced.** In particular, we apply SymILO in the training stage to modify labels by introducing permutations $\\pi^v$ as decisions in the loss minimization $\\min\_{\\theta,\\pi^v\_i}\\ell(f\_\\theta(\\mathcal{A}\_i,\\pi^v\_i(\\bar{x}\_i)))$, where the subscript $i$ indicates the $i$-th sample. The condition in this principle is approximately enforced by reducing the loss since violation will lead to la arger loss as motivated by Proposition 2 of the manuscript.
>
>
> Regarding the terminology, we believe terms like "approximate" and "relaxed" are both suitable to describe the approximate enforcement of isomorphic consistency, but the term "relaxed equivariance," as defined in [2], would not be appropriate in this context.
>
> >### **W3. About the experiments.**
> >#### **Finally, as also noted under questions, there seem to be weaknesses with the experiments — namely, the choice of loss function, and the premise/lack of comparison to non-ML baselines.**
>
> Thank you for your comments. Please refer to the responses to W2 and Q1.

---

> ### Author Response · Authors · 2024-11-25
> **Rebuttal Part III**
>
> >### **Q1. Value of ML versus classical ILP method.**
> >#### **Q1.1. what value does machine learning add to these problems? For example, are the trained models’ forward passes on test data supposed to be faster than the solver?**
>
> In classical ILP methods, the primary evaluation metric is typically the efficiency in producing feasible solutions with good objective values. In contrast, **ML-based methods aim to generate feasible solutions with better objective values more efficiently**, particularly by leveraging learned patterns in the data.
>
> To demonstrate the value of ML, we report the average objective values versus solving time (100–600s) when using predictions from the "augmentation" (Orbit+) model as a starting point for CPLEX. The BPP dataset is excluded from the analysis since those instances can be solved in just a few seconds.
>
> - **Dataset BIP**
>
> |   | 100s | 200s | 300s | 400s | 500s | 600s |
> |-|:-:|:-:|:-:|:-:|:-:|:---:|
> | CPLEX     | 16.9 | 16.2 | 15.8 | 15.5 | 15.3 | 15.2 |
> | CPLEX + "Orbit+"     | **15.5** | **15.1** | **14.9** | **14.8** | **14.7** | **14.6** |
>
> - **Dataset SMSP**
>
> |      | 100s | 200s | 300s | 400s | 500s | 600s |
> |---|:--:|:--:|:-:|:-:|:--:|:--:|
> | CPLEX     | 37.8 | 25.2 | 13.6 | 12.1 | 11.3 | 11.0 |
> | CPLEX + "Orbit+"         | **13.7** | **13.1** | **12.3** | **11.6** | **11.1** | **10.7** |
>
> From the results, we observe that **our augmentation scheme produces better objective values with less computational time**.
>
>
> >#### **Q1.2. does this take into account the time to detect the symmetry of the input ILP?**
>
> Yes, the results reported in Q1.1 include the time spent on symmetry detection. Additionally, we provide the average time cost of symmetry detection for our method across three datasets, with the size of the detected group $G$ represented by its logarithmic value $\\log\_{10} |G|$ as follows.
>
> |   |  BPP |  BIP |  SMSP |
> |:--:|:--:|:--:|:---:|
> | # of Var. |  420 | 1083  | 23000 |
> | $\\log\_{10} \|G\|$ |  6.9 |  6.6 | 213.1 |
> |    time(s)   | 0.05 | 0.06 |  6.11 |
>
> >#### **Q1.3. Does the noted symmetry-breaking problem arise when ML is used to predict branching decisions or node selections?**
>
> Yes, this issue also arises in ML-based tasks such as predicting branching decisions and node selections, as symmetric components (inputs) are present in these tasks as well.
> - For instance, consider variable branching involving two symmetric binary variables, $x\_1$ and $x\_2$. When branching on $x\_1$, two nodes are generated:
> $N\_{x\_1} = \\{ \\text{Node}(x\_1 = 0), \\text{Node}(x\_1 = 1) \\}$
> Similarly, branching on \\(x\_2\\) produces:
> $N\_{x\_2} = \\{ \\text{Node}(x\_2 = 0), \\text{Node}(x\_2 = 1) \\}$
> Crucially, $N\_1$ and $N\_2$ differ only in the variable names. If the model disregards these "name" distinctions—such as by applying permutation invariance—the outputs for $N\_{x\_1}$ and $N\_{x\_2}$ will be identical. This leads to an "inability to distinguish" between the two symmetric variables, potentially reducing the effectiveness of the branching strategy in exploring the solution space efficiently.
> - Similarly, for the node selection task, $\\text{Node}(x\_1=0)$ and $\\text{Node}(x\_2=0)$ represent symmetric components.
>
> >### **Q2. Choice of loss function in experiments.**
>
> Please refer to W2.
>
> >### **Q3. Motivation for “isomorphic consistency” principle**
>
> Please refer to W2.
>
> >### **Q4. Practicalities of the experiments**
> >#### **Q4.1. Line 468 states that multiple augmentations need to be drawn per sample. Is this done for baselines too?**
>
> No, the augmentation in the test stage is applied only once, as we found that sampling multiple times did not improve the average performance compared to sampling just once. This strategy is used consistently across all baselines as well as our methods.
>
> >#### **Q4.2 Also, is there an ablation result for the importance of using SymILO to enforce isomorphic consistency?**
>
>
> Thanks for raising this. Below is the validation loss ($\\downarrow$) of different models without (w/o) and with (w/) SymILO. The models trained with SymILO consistently and significantly perform better than others. Note that the performances of the "w/o" ones have slight differences, with variations only appearing beyond the third decimal place.
>
>
> |  Dataset | SymILO |    BPP   |    BIP   |   SMSP   |
> |:--------:|:------:|:--------:|:--------:|:--------:|
> |  No-Aug. |w/o |   0.40   |   0.34   |   0.94   |
> |  No-Aug. |  **w/** | **0.40** | **0.34** | **0.94** |
> |  Uniform |  w/o |   0.40   |   0.34   |   0.94   |
> |  Uniform |  **w/** | **0.27** | **0.31** | **0.72** |
> | Position |  w/o |   0.40   |   0.34   |   0.94   |
> | Position |  **w/** | **0.23** | **0.30** | **0.72** |
> |   Orbit  |  w/o |   0.40   |   0.34   |   0.94   |
> |   Orbit  |  **w/** | **0.22** | **0.29** | **0.59** |
> |  Orbit+  |  w/o  |   0.40   |   0.34   |   0.94   |
> |  Orbit+  |  **w/** | **0.21** | **0.28** | **0.58** |

---

> ### Author Response · Authors · 2024-11-25
> **Rebuttal Part IV**
>
> >### **Q5. Minor clarification about formulation symmetry definition**
> >#### **Q5.1 One way of achieving this is if $Ag(x)=Ax$, indeed this is what I would have expected as the definition. Is this more accurate?**
>
> In fact, there are multiple definitions of symmetries in the ILP literature, and permuting variables is also a frequently used one. Please refer to Section 17.2 of [15] for more details.
>
> >#### **Q5.2. around lines 228-230, the paper says “…the approach did not exploit the underlying symmetry properties, leading to suboptimal performance on ILPs with strong symmetries”. What does this mean? The paper would be improved by making this claim precise, and including references to evidence.**
>
> This statement suggests that while machine learning-based algorithms for integer linear programming (ILPs) may perform well on problems with minimal symmetries, they tend to underperform or show suboptimal results on problems with significant symmetries. The primary reason for this discrepancy is that these algorithms do not exploit the symmetry inherent in the problem structure, which hinders their ability to solve the problem efficiently when symmetries are pronounced. The claim is supported by experimental evidence from [16], which demonstrates that symmetry-agnostic methods perform notably worse compared to symmetry-aware methods.
>
> >### **Q6  Why not use MIPLIB**
>
> >#### **Q6.1 Why not use MIPLIB, the dataset cited in the first paragraph of the paper, in the experiments?**
>
> The MIPLIB dataset includes instances from a wide range of real-world applications, making it valuable for representing the diversity of real-world problems. However, as noted in [12], it is not ideal for machine learning applications. The dataset is described in [12] as "**a heterogeneous dataset combining many applications that a priori would not be considered similar enough to have sufficient shared structure for learning to exploit**." This lack of shared structure limits the ability of machine learning models to effectively generalize across the instances, reducing its utility for learning-based methods.
>
> >#### **Q6.2 For the datasets used in experiments, what fraction of inputs exhibit no symmetric variables at all?**
>
> The fraction is $0$, as all the problems considered in our experiments—BBP, BIP, and SMSP—exhibit significant inherent symmetries.
>
> ---
> ### **Typos and writing suggestions**
>
> We sincerely thank you for the detailed review. We are currently revising our manuscript and will upload the updated version as soon as possible.
>
> ---
> ### **References**
>
> [1] Smidt, T. E., Geiger, M., and Miller, B. K. Finding symmetry breaking order parameters with Euclidean neural networks. Phys. Rev. Research, 3: L012002, Jan 2021. doi: 10.1103/PhysRevResearch
>
> [2] Kaba, S.-O. and Ravanbakhsh, S. Symmetry breaking and equivariant neural networks. In Symmetry and Geometry in Neural Representations Workshop, NeurIPS, 2023.
>
> [3] Xie, Y. and Smidt, T. Equivariant symmetry breaking sets. TMLR 2024.
>
> [4] Hannah Lawrence, Vasco Portilheiro, Yan Zhang, and Sekou-Oumar Kaba. Improving equivariant networks with probabilistic symmetry breaking. In ICML 2024 Workshop on Geometry-grounded Representation Learning and Generative Modeling, 2024.
>
> [5] Zhang, Y., Zhang, D. W., Lacoste-Julien, S., Burghouts, G. J., and Snoek, C. G. M. Multiset-equivariant set prediction with approximate implicit differentiation. In International Conference on Learning Representations, 2022.
>
> [6] Morris, M., Grau, B. C., & Horrocks, I. (2024). Orbit-equivariant graph neural networks. ICLR 2024.
>
> [7] Pfetsch M E, Rehn T. A computational comparison of symmetry handling methods for mixed integer programs[J]. Mathematical Programming Computation, 2019, 11: 37-93.
>
> [8] Park S, Van Hentenryck P. Self-supervised primal-dual learning for constrained optimization[C]  AAAI 2023.
>
> [9] Schuetz M J A, Brubaker J K, Katzgraber H G. Combinatorial optimization with physics-inspired graph neural networks[J]. Nature Machine Intelligence, 2022, 4(4): 367-377.
>
> [10] Min Y, Bai Y, Gomes C P. Unsupervised learning for solving the traveling salesman problem[J].  NeurIPS 2024.
>
> [11] Ding J Y, Zhang C, Shen L, et al. Accelerating primal solution findings for mixed integer programs based on solution prediction[C]// AAAI  2020
>
> [12] Nair V, Bartunov S, Gimeno F, et al. Solving mixed integer programs using neural networks[J]. arXiv preprint arXiv:2012.13349, 2020.
>
> [13] Khalil E B, Morris C, Lodi A. Mip-gnn: A data-driven framework for guiding combinatorial solvers[C] AAAI 2022.
>
> [14] Han Q, Yang L, Chen Q, et al. A gnn-guided predict-and-search framework for mixed-integer linear programming[J]. ICLR 2023.
>
> [15] Margot F. Symmetry in integer linear programming[J]. 50 Years of Integer Programming 1958-2008: From the Early Years to the State-of-the-Art, 2009: 647-686.
>
> [16] Chen Q, Zhang T, Yang L, et al. SymILO: A SymmetryNeurIPS-Aware Learning Framework for Integer Linear Optimization[J]. NeurIPS 2024.

---

> > ### Comment · Reviewer_GLPM · 2024-11-27
> > **Thanks for the responses!**
> >
> > Thank you very much to the authors for their detailed response. My responses are in order below.
> >
> > 1. *Concerns about novelty*: The authors said they “added a paragraph in the revised manuscript to acknowledge and cite these relevant works”. Where was this done? I did not see it anywhere, nor were any of the references I added in the related works.
> > 2. *Novelty in the context of ILPs*: Perhaps I am misunderstanding — how is this work the first to identify the indistinguishability issue in ILPs, when it was observed and addressed by many of the papers the authors cited themselves, such as “Symmetry in Integer Linear Programming” by Francois Margot, the SymILO paper, Chen 2022’s concept of “foldable” ILPs (which is very related), as well as positional encoding approaches by Han et al 2023 and Chen et al 2024? I agree that these prior works should have referred to the rich symmetry-breaking literature in sets and physical applications too, but wouldn’t count this as one of this submission’s novelties, especially considering it’s not even in the manuscript yet.
> > 3. *Special problem structures in ILPs*: the addition of augmented features to only the variable nodes is a fair point, as well as the structures adopted in the “Orbit+” approach.
> > 4. *Comparison to Orbit-Inv-GNN*: this is a reasonable comparison, and would be worth exploring more in the paper — for instance, it is not self-explanatory how overfitting works here. Can the authors expand on this more?
> > Also, since 2x and 8x sampling are more expensive, this is not necessarily a fair comparison — what about 1x-train vs 1x-valid, which are randomized, as compared to the existing Det-train and Det-valid rows? Also, I would encourage the authors to add a reference to this work in their manuscript, and the discussion from the rebuttal.
> > 5. *Orbit-invariant loss*: I see the author’s points about using a supervised loss due to integrality constraints, but I still don’t see how an orbit-invariant loss function is out of scope — one could always take the existing loss function, but minimize it over the automorphism group of the variable nodes in the graph (ie, using a heuristic to find an optimal permutation to match the network’s output to the supervised output). An “orbit-invariant” loss function seems to me much more natural than isomorphic consistency, since the whole principle of the setup is that some of the variables are symmetric and “equivalent” to each other. I can’t quite understand the W2.2 response, and whether the minimization over $\pi_i^v* constitutes an orbit or even group-invariant loss function. (Incidentally, I am still fairly confident that relaxed equivariance is a relevant concept here — it is a general concept related to symmetry-breaking in a deterministic way, of which this whole ILP setting is a special case. Perhaps the authors can justify more why it is, in their opinion, inappropriate?)
> > 6. *Isomorphic consistency W2.2*: Unfortunately, I have really tried to understand isomorphic consistency from both the manuscript and the authors’ rebuttal, but I think I still find it pretty unclear what they mean. In particular, I still don’t understand the “theoretically it is achievable” explanation. Isn’t it a property of the dataset itself? I.e. isn’t it possible that the dataset labels make isomorphic consistency impossible? And where are $z$ and $z’$ in this explanation? Also, in the “practically, it can be approximately enforced” section, where is the ground truth label in the expression of the loss function?
> > 7. *ML vs ILP, and symmetry group computation time*: Thanks for the timing tests. These motivate the work and would be crucial additions to the paper.
> > 8. *symmetry-breaking in branching prediction*: great, makes sense.
> > 9. *all subsequent responses*: these were all satisfactory, thanks.
> >
> > In sum, I think the authors’ new experiments have made their work stronger, but I am keeping my original rating at the moment because (1) the novelty is still kind of limited relative to prior work, and (2) the presentation of isomorphic consistency, the relation to prior symmetry-breaking work (which is missing from the PDF), and the relation to orbit graph networks (in terms of both the “overfitting” explanation, and modifications to the actual manuscript) is lacking. To me, the explanations of isomorphic consistency and (less importantly, since this was in the rebuttal and not the manuscript) the “randomness vs deterministic” comparison to orbit GNNs were particularly confusing, yet I suspect related to other concepts in the symmetry-breaking literature — and I would really like to understand them in order to evaluate them fairly. I really think this work has promise, and would probably increase my rating if all aspects of (2) were satisfactorily addressed.

---

> ### Author Response · Authors · 2024-11-30
> **Reply Part I**
>
> First and foremost, we would like to express our sincere gratitude for your thoughtful responses. Below, we offer further clarifications on the questions that remain unaddressed.
>
> >### **Q1. Concerns about novelty**
> >**About revision: ... Where was this done? I did not see it anywhere, nor were any of the references I added in the related works.**
>
> We apologize for the delay in submitting the revised manuscript, which was submitted one day later than our previous response. The updated version is now available, and the revisions made in this version include the following:
> - Adding a paragraph in the introduction (lines 71-85) to acknowledge and cite the relevant works.
> - Rewriting the motivation part (lines 86-93) to provide a more precise and concrete explanation, incorporating the newly cited works.
> - Clarifying that the core ideas of "distinguishability" and "augmentation parsimony" are derived from the existing works (lines 245-256, and 273-275 in particular).
> - Adding experimental results in Appendix A.6 of the manuscript.
>
> >### **Q2. Novelty in the context of ILPs.**
> >**About novelty: ... how is this work the first to identify the indistinguishability issue in ILPs, ...**
>
> Thank you for pointing this out. We apologize for the inaccuracy in our original statement. A more precise version would be: "Our work is the first to address the issue of indistinguishability within the ML-based ILP literature by leveraging the intrinsic properties of ILPs." We have incorporated similar clarification in the motivation part of the updated manuscript.
>
>
> >### **Q4. Comparison to Orbit-Inv-GNN**
>
> Thank you for your valuable questions. Below, we provide a more detailed explanation of the experiment. Our original description was not precise. We now restate the experiment with more clarity. The updated results for the loss ($\\downarrow$) are as follows:
> |training on|evaluating on| |||epoch|||
> |:---:|:--:|--|:--:|:-:|:-:|:----:|:--:|
> ||||10|30| 50|70|90|
> | train. set (Det)| train. set (Det) | | 0.24 |0.22|0.21 | 0.20 | 0.18 |
> | train. set (Det)| **valid. set (Det)** | | 0.24 | 0.23 |0.24| 0.24 | 0.25 |
> |  train. set (1x)|train. set (1x) | | 0.24 | 0.23 |0.22| 0.20 | 0.18 |
> |  train. set (1x)|**valid. set (1x)** || 0.24 | 0.23 |0.23 | 0.24 | 0.25 |
> |  train. set (2x)|train. set (2x) || 0.24 | 0.23 | 0.22| 0.22 | 0.21 |
> |  train. set (2x)|**valid. set (1x)** ||0.25| 0.23 |0.23| 0.23 | 0.23 |
> |train. set (8x)|train. set (8x)||0.24| 0.23 |  0.22 | 0.21 |0.21|
> |train. set (8x) |**valid. set (1x)**|| 0.24 | 0.23 |  0.22 | 0.22 | 0.22 |
>
> where
> - "Det" refers to the deterministic augmentation strategy (as in Orbit-Inv-GNN), where each instance gets a fixed augmented feature.
> - "$k$x" refers to the random augmentation strategy, where $k$ samples are drawn for each instance (as detailed in Algorithm 1). If the dataset has $n$ instances, "$k$x" means $k \\cdot n$ augmented samples are generated.
> - All validation sets with different augmentation strategies ("Det" and "1x") have the same size.
>
>
> >#### **Q4.1. this is a reasonable comparison, and would be worth exploring more in the paper — for instance, it is not self-explanatory how overfitting works here. Can the authors expand on this more?**
>
> Thank you for raising this valuable question. Below are additional clarifications:
>
> The deterministic approach generates augmented features by assigning IDs $\\{1,\\dots,m\\}$ to variable nodes in the same orbit in a fixed order (e.g., variable with smaller index get smaller ID). In contrast, the random strategy samples multiple shuffled orderings of these IDs, producing different feature augmentations for each instance. This randomness increases the size and diversity of the training set, which is key to reducing overfitting.
>
> >#### **Q4.2. Also, since 2x and 8x sampling are more expensive, this is not necessarily a fair comparison — what about 1x-train vs 1x-valid, which are randomized, as compared to the existing Det-train and Det-valid rows?**
>
> - **Regarding computational cost**: **The comparison at the inference (validation) stage is fair.** While 8x and 2x sampling increase the size of the training set, leading to higher computational costs during training, the validation stage uses only 1x sampling. As a result, the inference cost remains the same for both the deterministic and randomized strategies.
>
> - **Regarding "1x-train vs 1x-valid"**: As shown in the results, the 1x-train strategy with random sampling still suffers from significant overfitting, similar to the deterministic strategy.
>
> >#### **Q4.3. Also, I would encourage the authors to add a reference to this work in their manuscript, and the discussion from the rebuttal.**
>
> - Reference: The citation has been added in the introduction (line 78).
> - Orbit-Inv-GNN comparison: We included this discussion in our local revised version, but the submission portal closed before we could update the manuscript. It will be included in the next version.

---

> ### Author Response · Authors · 2024-11-30
> **Reply Part II**
>
> >### **Q5. Orbit-invariant loss**
> >#### **Q5.1. ... one could always take the existing loss function, but minimize it over the automorphism group of the variable nodes in the graph ...**
>
> Thank you very much for your helpful clarification regarding the concept of "orbit-invariant loss". After reflecting on your explanation, we now have a clearer understanding of its meaning.
>
> There are two potential interpretations of an "orbit-invariant" loss that we are now aware of:
> - **Loss based on ILP objective and constraint violations:** We didn't use it because of the integrality constraints, as explained in our previous responses. (Initially, we misunderstood the term "orbit-invariant," believing it referred solely to this interpretation. However, we have since realized that it carries a second meaning.)
> - **A loss function minimizing over the automorphism group:** **This is exactly what we are doing in our approach (we now realize it)!** Specifically, we employ the SymILO framework, which incorporates a loss function of the form: $\\min\_{\\theta,\\pi\_i\\in G\_i}\\frac{1}{N}\\sum\_{i=1}^N\\ell\\left(f\_\\theta(\\tilde{\\mathcal{A}}\_i),\\pi^v\_i(\\bar{x}\_i)\\right)$ where $G\_i$ represents the formulation symmetries (automorphism group) of instance $\\mathcal{A}\_i$, and $\\pi\_i$ is a permutation in the automorphism group being optimized to match the label $\\bar{x}\_i$ with the prediction $f\_\\theta(\\tilde{\\mathcal{A}}\_i)$. This loss function is indeed "orbit-equivariant" and, as you noted, "group-invariant."
>
>
> >#### **Q5.2. An “orbit-invariant” loss function seems to me much more natural than isomorphic consistency, since the whole principle of the setup is that some of the variables are symmetric and “equivalent” to each other.**
>
> Thank you for your insightful comment.  As discussed in Q5.1, we now realize that we are indeed using an orbit-invariant loss function in our approach.
>
> In addition, I would like to further clarify the relationship between isomorphic consistency and orbit-invariant loss:
> - **They are distinct concepts**: Isomorphic consistency refers to a guiding principle that imposes a condition on the training samples, which can enhance model performance. In contrast, an orbit-invariant loss function is a specific technique aimed at approximately (and partially) enforcing isomorphic consistency.
> - **Orbit-invariant loss cannot fully replace isomorphic consistency:** While the orbit-invariant loss is a useful tool, it does not encapsulate all aspects of isomorphic consistency. Other methods can also be employed to enforce isomorphic consistency. Please refer to our responses to Q6.
>
>
> >#### **Q5.3. I can’t quite understand the W2.2 response, and whether the minimization over $\\pi\_i^v constitutes an orbit or even group-invariant loss function.**
>
> Thank you for your comment, we apologize for any confusion caused.
> - For a more detailed explanation of the W2.2, please refer to our response in Q6, where we provide a detailed discussion.
> - Regarding the minimization over $\\pi\_i^v$: Yes, it constitutes an orbit and group-invariant loss function, as mentioned in the responses to Q5.1.

---

> > ### Author Response · Authors · 2024-11-30
> > **Reply Part III**
> >
> > >#### **Q5.4. (Incidentally, I am still fairly confident that relaxed equivariance is a relevant concept here ... Perhaps the authors can justify more why it is, in their opinion, inappropriate?)**
> >
> > Thank you for raising this insightful question. We appreciate the opportunity to provide further clarification on this matter.
> >
> > We acknowledge that the concept of relaxed equivariance presented in [1] is indeed relevant; however, due to certain differences, it is not directly applicable to our settings. Furthermore, we argue that the notion of **"equivariance/invariance in the ground truth function" provides a more precise interpretation of isomorphic consistency**.
> >
> > 1. **Why relaxed equivariance is inappropriate.**
> >     - To our knowledge, relaxed equivariance (as defined in [1]) is a property of function, especially when "symmetry breaking" is a part of the function itself. In contrast, our approach introduces symmetry breaking during the input preprocessing stage, where features are augmented into a bipartite graph. This preprocessing step breaks the symmetry prior to inputting the graph into the GNN model.
> >     - While one could consider this preprocessing step as part of the GNN function, the concept of relaxed equivariance may still be unsuitable for our approach, or would require further extension, due to the inherent randomness introduced by the augmented features.
> >
> > Nevertheless, we acknowledge that relaxed equivariance has deeper connections to our method (and to [2]), and we believe it warrants further exploration in future work.
> >
> > 2. **Isomorphic consistency as "equivariance/invariance" in the target function.**
> >
> >
> > In supervised learning, the training set is designed to reflect the target (or groundtruth) function. Each training sample corresponds to a point in this target function, meaning that **imposing isomorphic consistency on the training samples effectively imposes it on the target function as well.** Base on this, we define isomorphic consistency directly on the target function itself. Let $f^*$ denote the target function, which takes a bipartite graph with augmented features as input and outputs a prediction for the variable nodes.
> > For brevity, let $\\tilde{A}$ represent the bipartite graph with augmented features (as defined in Section 4.1 of the manuscript). The isomorphic consistency of the target function can be formally defined as follows.
> >
> > **Definition 4: (Isomorphic consistency of the target function)** If $(\\mathcal{\\tilde{A}},\\bar{x})$ and $(\\mathcal{\\tilde{A}}',\\bar{x}')$ are two points on $f^*$, then $\\forall \\pi \\in S\_n, \\sigma \\in S\_m, \\pi^c(\\sigma^r(\\tilde{\\mathcal{A}}))=\\tilde{\\mathcal{A}}' \\implies \\pi^v(\\bar{x})=\\bar{x}'$.
> >
> > Based on the isomorphic consistency of the target function, the following proposition holds.
> >
> > **Proposition 3:** If $f^*$ satisfies the isomorphic consistency, then for any $\\tilde{\\mathcal{A}}$, we have $f^*(\\pi^c(\\tilde{\\mathcal{A}}))=\\pi^v(f^*(\\tilde{\\mathcal{A}}))$ and $f^*(\\sigma^r(\\tilde{\\mathcal{A}}))=f^*(\\tilde{\\mathcal{A}})$.
> >
> > **Proof:** For arbitrary $\\tilde{\\mathcal{A}}$, $\\pi$ and $\\sigma$, let $\\tilde{\\mathcal{A}}'=\\pi^c(\\sigma^r(\\tilde{\\mathcal{A}}))$:
> > - First, $f^*(\\pi^c(\\sigma^r(\\tilde{\\mathcal{A}})))=f^*(\\tilde{\\mathcal{A}}')=\\bar{x}'=\\pi^v(\\bar{x})=\\pi^v(f^*(\\tilde{\\mathcal{A}}))$.
> >
> > Since $f^*(\\pi^c(\\sigma^r(\\tilde{\\mathcal{A}})))=\\pi^v(f^*(\\tilde{\\mathcal{A}}))$ holds for arbitrary $\\pi$ and $\\sigma$, we can assume they are identity permutations, respectively:
> > - When $\\pi$ is the identity permutation, $f^*(\\sigma^r(\\tilde{\\mathcal{A}}))=f^*(\\pi^c(\\sigma^r(\\tilde{\\mathcal{A}})))=f^*(\\tilde{\\mathcal{A}})$ (i.e., permutation invariance).
> > - When $\\sigma$ is the identity permutation, $f^*(\\pi^c(\\tilde{\\mathcal{A}}))=f^*(\\pi^c(\\sigma^r(\\tilde{\\mathcal{A}})))=\\pi^v(f^*(\\tilde{\\mathcal{A}}))$ (i.e., permutation equivariance).
> >
> > From this, we find that isomorphic consistency enforces permutation invariance on the constraint nodes and permutation equivariance on the variable nodes. These two properties are exactly what we assume in Assumption 1 of the manuscript for the GNN function.
> >
> > In supervised learning, where the goal is to predict an ILP solution, it is natural for the target function to exhibit these two properties. However, it is important to note that achieving such consistency in practice can be challenging. Please refer to our response to Q6 for more clarifications.

---

> ### Author Response · Authors · 2024-11-30
> **Reply Part IV**
>
> >### **Q6. Isomorphic consistency W2.2**
> >#### **Q6.1. Unfortunately, I have really tried to understand isomorphic consistency from both the manuscript and the author’s rebuttal, but I think I still find it pretty unclear what they mean.**
>
> Thank you for your continued efforts in seeking a deeper understanding of the concept of isomorphic consistency. We really appreciate it, and we apologize for any confusion. To provide further clarity, we have offered a more detailed explanation in our response to Q5.4.
>
> In summary, **isomorphic consistency refers to an "equivariance/invariance" condition applied to the target function (the ground truth function), distinguishing it from existing literature, which typically focuses on such conditions for the model function.** We hope this clarification helps provide a clearer understanding of the role of isomorphic consistency within our framework.
>
> Should any aspects remain unclear, we would be happy to address any further questions in more detail.
>
> >#### **Q6.2. In particular, I still don’t understand the “theoretically it is achievable” explanation. Isn’t it a property of the dataset itself? I.e. isn’t it possible that the dataset labels make isomorphic consistency impossible?**
>
> Thank you for your insightful comment. We apologize for any confusion caused by our previous explanation. The issue stems from an implicit assumption in our work that was not clearly articulated, and we appreciate the opportunity to clarify this matter.
>
> In ILP research, datasets typically consist of ILP instances, but the corresponding solutions are not provided directly. Instead, these solutions are usually generated by an ILP solver, which identifies and stores feasible (or optimal) solutions during the solving process. In our setting, we assume that any of these optimal solutions can be selected as the label, with the most common choice being the solution that yields the best objective value. This leads to the following implicit assumption:
>
> - **Implicit assumption (labels are changeable):** For any ILP instance, we assume that the label can be chosen from any of its optimal solutions (e.g., one of several equivalent optimal solutions).
>
> With this assumption in mind, we can revisit the notion of isomorphic consistency.
>
> Consider two isomorphic instances $\\mathcal{A}$ and $\\mathcal{A}'$, which are related by an isomorphism $(\\pi,\\sigma)$ such that $\\pi^c(\\sigma^r(\\mathcal{A})) = \\mathcal{A}'$. In other words, these two instances are essentially the same problem, but with different orderings of variables and constraints. We have elaborated on this point further in Appendix A.2 of the manuscript.
>
> Now, for the augmented features $z$ and $z'$ associated with $\\mathcal{A}$ and $\\mathcal{A}'$, respectively, there are two ways to ensure isomorphic consistency:
>
> - **Ensuring $\\pi^v(z) \\neq z'$:** In this case, the isomorphic consistency condition $\\pi^v(z) = z' \\implies \\pi^v(\\bar{x})=\\bar{x}$ holds. This can be achieved by re-sampling, but has its own issue as mentioned in Section 4.3.1 of our manuscript.
> - **Ensuring $\\pi^v(\\bar{x})=\\bar{x}'$:** Under this condition, isomorphic consistency holds **regardless of whether $\\pi^v(z) = z'$**. This approach relies on our implicit assumption that we can select $\\bar{x}$ and $\\bar{x}'$ from the optimal solutions of $\\tilde{\\mathcal{A}}$ and $\\tilde{\\mathcal{A}}'$, respectively. Since these two instances are isomorphic w.r.t. $\\pi$, there must exist two solutions satisfying $\\pi^v(\\bar{x})=\\bar{x}'$.
>
> Based on the above two ways, we claimed that "theoretically it is achievable".
>
>
>
> >#### **Q6.3. And where are $z$ and $z'$ in this explanation?**
>
> In our approach, we employ SymILO, which belongs to the second way of ensuring isomorphic consistency mentioned in Q6.2. This approach ensures isomorphic consistency without the need to consider $z$ and $z'$. Therefore, we didn't mention $z$ and $z'$ in particular.
>
> >#### **Q6.4. Also, in the “practically, it can be approximately enforced” section, where is the ground truth label in the expression of the loss function**
>
> In response to Q5.1, we have an improved explanation of the loss function, please refer to it and let us know if it is not clear.
>
>
> >### **Q7. ML vs ILP, and symmetry group computation time**
> >#### **Thanks for the timing tests. These motivate the work and would be crucial additions to the paper.**
>
> Thank you for your valuable suggestion. We have added them in Appendix 6 of the revised manuscript.
>
>
> ---
> ### **Reference**
>
> [1] Kaba, S.-O. and Ravanbakhsh, S. Symmetry breaking and equivariant neural networks. In Symmetry and Geometry in Neural Representations Workshop, NeurIPS, 2023.
>
> [2] Morris, M., Grau, B. C., & Horrocks, I. (2024). Orbit-equivariant graph neural networks. ICLR 2024.

---

> > ### Comment · Reviewer_GLPM · 2024-12-02
> > **Thanks - increasing my score**
> >
> > Thanks to the authors for their detailed engagement with my comments. I have read their responses and looked at the updated manuscript, and am now happy to recommend the paper for acceptance. I have a somewhat improved understanding of isomorphic consistency now, although I still feel there must be a clearer way of saying it (perhaps for instance by stating things explicitly in terms of the stabilizer of $\mathcal{A}$, or by first articulating what it says when the input does not have a self-symmetry). I also apologize for not precisely defining "orbit-invariant loss" before.
> >
> > Just a minor note: although the subtitle of Assumption 1 was fixed to "equivariance", the text inside the assumption still says "equivalent".

---

> > > ### Author Response · Authors · 2024-12-02
> > >
> > > We would like to express our heartfelt gratitude for your thoughtful and detailed feedback on our manuscript. Your comments have significantly deepened our understanding of "symmetry breaking" in ML, and your suggestions have greatly contributed to improving the quality of our paper. We sincerely appreciate the time and effort you've invested in reviewing our work.
> > >
> > > Regarding your suggestion on the clarification of "isomorphic consistency," we will follow your advice and provide a more detailed explanation in the revised manuscript. Specifically, we will articulate it more clearly, potentially by discussing the stabilizer of the input or explicitly describing what happens when the input does not have a self-symmetry. Additionally, we will correct some of the typos and imprecise expressions that you pointed out (e.g., equivariance), as well as incorporate the experiments and analysis that we were unfortunately unable to update during the rebuttal phase.
> > >
> > > **On a separate note, we noticed that the score for the official comment has not yet been updated in the system. We fully understand that such things can be overlooked, but we wanted to kindly bring it to your attention.**
> > >
> > > Once again, we are deeply grateful for your invaluable feedback and for the constructive dialogue we’ve had throughout this process. It has been a pleasure working through these points with you, and we truly appreciate your contributions to improving our manuscript.

---

> > > > ### Comment · Reviewer_GLPM · 2024-12-03
> > > > **Done!**
> > > >
> > > > Thanks for pointing that out. Updated now! I also enjoyed the discussion, and thank you for engaging thoughtfully with feedback.

---

### Official Review · Reviewer_zedB · 2024-11-06

**Soundness:** 2
**Presentation:** 2
**Contribution:** 2
**Rating:** 5
**Confidence:** 2

**Summary:**

The authors study the problem of solving Integer Linear Programs (ILPs) with symmetry among variables. They first show that if a permutation from the set of all variable permutations is a formulation symmetry of the ILP, then under an assumption of permutation equivalence and invariance for the GNN, the network cannot predict the optimal solution of the ILP.

To address this, they propose a feature augmentation algorithm that assigns unique augmented features to each orbit, sampling a distinct feature value within an orbit without replacement. They compare their methods against previously proposed augmentation schemes empirically and based on some principles for three ILP benchmark problems.

**Strengths:**

Their method appears to perform better in terms of their proposed metric than existing methods on certain tasks. They also introduce the problem clearly, making it easily understandable for someone outside the field, while establishing a good motivation through negative results about GNNs and formulation symmetries.

**Weaknesses:**

They don't discuss any limitation of their orbit-based augmentation, making their method's application scope appear narrow restricting to ILPs with formulation symmetry.

Additionally, the approach relies on detecting symmetry groups and orbits, which as they note may be computationally expensive.  It would also be interesting to see how their method performs for different evaluation metrics (maybe beyond $\ell_1$ distances).

As this area is new to me, their contributions do not seem sufficiently novel in terms of the algorithm, and the experiments provided also seem limited and hence I recommend a reject but with a confidence score of 2.

**Questions:**

Could the authors comment on the computational complexity of detecting symmetries and how the established methods help handle it?

---

> ### Author Response · Authors · 2024-11-25
> **Rebuttal Part I**
>
> >### **W1. Lack of discussion about the limitation**
> >#### **W1.1 They don't discuss any limitation of their orbit-based augmentation**
>
> Thank you for pointing out this. In response, we have added a limitations section in the revised version of the manuscript, which is also summarized below:
>
> First, our approach is specifically tailored for ILPs with formulation symmetry, and it is currently unknown whether it can effectively be applied to problems of other classes. Second, the principle of isomorphic consistency, which underpins our method, is primarily applicable to supervised learning tasks where multiple label choices exist. These limitations highlight areas for further exploration and potential extension of our approach.
>
>
> >#### **W1.2 ... making their method's application scope appear narrow restricting to ILPs with formulation symmetry.**
>
> Thank you for your insightful comment. Our method is tailored for ILPs with symmetry, and we have not yet explored its application to other types of problems in this work. However, we hope that our work can inspire further research and development within the optimization and machine-learning communities, leading to potential extensions of our approach.
>
> It is also important to note that **ILPs with symmetry are commonly encountered in practical applications**. For instance, in MIPLIB [1], a well-known collection of real-world ILPs from diverse fields, nearly 32% of instances exhibit certain formulation symmetries. This highlights the relevance and applicability of our method to a broad range of problems.
>
> >### **W2. Complexity of symmetry detection**
> >#### **The approach relies on detecting symmetry groups and orbits, which as they note may be computationally expensive.**
>
> Thank you for your valuable comment. Calculating the full symmetry group of an ILP can indeed be challenging and potentially NP-hard. However, detecting a subgroup, as discussed in Section 17.3 of [2], is more practical and tractable. There are also well-established tools, such as Bliss [4], Nauty, and Traces [5], which integrate efficient group detection algorithms that can heuristically identify symmetry subgroups in a relatively short amount of time. To illustrate this, we report the average time taken to detect symmetry groups in the considered datasets using Bliss, with the size of the detected subgroup $G$ represented by its logarithmic value $\\log_{10} |G|$
>
> |   | BPP|BIP|SMSP|
> |:-:|:-:|:-:|:-:|
> | # of Var. |  420 | 1083  | 23000 |
> | $\\log_{10} \|G\|$ |  6.9 |  6.6 | 213.1 |
> | time(s)   | 0.05 | 0.06 |  6.11 |
>
> From the results, we observe that for smaller problems like BPP and BIP, symmetry detection requires negligible computational time. Even for larger problems, such as SMSP with 23,000 variables, symmetry group detection is still accomplished within a few seconds, demonstrating the feasibility of our approach even for complex instances.
>
>
> >### **W3. About evaluation metrics**
> >#### **It would also be interesting to see how their method performs for different evaluation metrics (maybe beyond $\\ell_1$ distances).**
>
> Thank you for your insightful comment. In response, we have supplemented our analysis with two additional evaluation metrics beyond $\\ell_1$ distances:
>
> 1. **Objective values ($\\downarrow$)**: We report the average objective value versus solving time (100–600s) when using the predictions from the "augmentation" (Orbit+) model and "no-augmentation" (No-Aug) one to enhance the ILP solver CPLEX with the fixing strategy from [3]. Note that the dataset BPP is excluded since those instances could solved in very few seconds.
>
>     - **Dataset BIP**
>
>     |  | 100s | 200s | 300s | 400s | 500s | 600s |
>     |-|:-:|:-:|:-:|:-:|:-:|:-:|
>     | No-Aug  | 16.9 | 16.2 | 15.8 | 15.5 | 15.3 | 15.2 |
>     | Orbit+ | **15.5** | **15.1** | **14.9** | **14.8** | **14.7** | **14.6** |
>
>     - **Dataset SMSP**
>
>     | | 100s | 200s | 300s | 400s | 500s | 600s |
>     |-|:-:|:-:|:-:|:-:|:-:|:-:|
>     | No-Aug| 37.8 | 25.2 | 13.6 | 12.1 | 11.3 | 11.0 |
>     | Orbit+| **13.7** | **13.1** | **12.3** | **11.6** | **11.1** | **10.7** |
>
>     From the results, we observe that **our augmentation scheme produces better objective values with less computational time**.
>
> 2. **Constraint violations ($\\downarrow$)**: We also report the total violation of model prediction $\\hat{x}$ with respect to the constraint $Ax \\leq b$, i.e., the summation of positive elements in $Ax - b$. The violation of predictions from different models is summarized below.
>
>     | |BIP|SMSP|
>     |:-:|:-:|:-:|
>     |Uniform|27.84|852.23|
>     |Position|22.12|933.76|
>     |Orbit|12.89|453.87|
>     |Orbit+|**3.68**|**329.74**|
>
> From the results, we find that **our methods (Orbit, Orbit+) produce predictions with significantly less constraint violation**, further demonstrating the effectiveness of our approach.
>
> We hope these additional metrics provide a more comprehensive evaluation of the performance of our method and address your suggestion.

---

> > ### Author Response · Authors · 2024-11-25
> > **Rebuttal Part II**
> >
> > >### **W4. About the novelty of the algorithm**
> > >#### **As this area is new to me, their contributions do not seem sufficiently novel in terms of the algorithm, and the experiments provided also seem limited and hence I recommend a reject but with a confidence score of 2.**
> >
> >
> > Thank you for your feedback. We understand that it might be challenging to assess a paper outside of one's area of expertise, and we appreciate your thoughtful comments. To provide more clarity on our contributions, we highlight the following points:
> >
> > 1. **Identification of the issue that GNNs are unable to distinguish symmetrical variables for ILPs**: We provide a theoretical insight into a significant challenge when applying Graph Neural Networks (GNNs) to predict solutions of ILPs with symmetry. To the best of our knowledge, this issue has not been adequately addressed in the existing ILP literature, and our paper brings attention to this gap.
> >
> > 2. **A feature augmentation scheme to address the issue**: In response to the identified issue, we propose a feature augmentation scheme and present three guiding principles for designing augmented features. Our **orbit-based feature augmentation approach**, developed based on these principles, **leads to a significant improvement** in model performance.
> >
> > **While the proposed algorithm is simple to implement, it is far from trivial.** The theoretical analysis involved, as well as the application of structural insights from ILPs, are non-trivial and require careful consideration to achieve effective results.
> >
> > We believe this paper addresses a critical gap at the intersection of GNNs and ILP, offering significant advancements in both theoretical understanding and practical applications
> >
> >
> > >### **Question**
> > >#### **Q1. Could the authors comment on the computational complexity of detecting symmetries...?**
> >
> > Thanks for raising this question. As we mentioned in our response to W2, calculating the full symmetry group of an ILP can indeed be computationally challenging and potentially NP-hard. However, detecting a symmetry subgroup heuristically is a much more practical and feasible approach. We leverage well-developed tools such as Bliss [4], Nauty, and Traces [5] to identify subgroups, which are computationally more efficient and applicable to larger ILP instances without incurring significant computational costs.
> >
> >
> > >#### **Q2. ... and how the established methods help handle it**
> >
> > We are not sure about what "it" refers to. If it refers to "detecting symmetry", then there might be some misunderstanding. The augmentation scheme does not help symmetry detecting, instead symmetry detection is one step producing the required elements -- orbits in our scheme. If not, then we kindly ask the reviewer to clarify this question a little so that we can address your concern.
> >
> >
> > ---
> > ### **References**
> >
> > [1] Gleixner A, Hendel G, Gamrath G, et al. MIPLIB 2017: data-driven compilation of the 6th mixed-integer programming library[J]. Mathematical Programming Computation, 2021, 13(3): 443-490.
> >
> > [2] Margot F. Symmetry in integer linear programming[J]. 50 Years of Integer Programming 1958-2008: From the Early Years to the State-of-the-Art, 2009: 647-686.
> >
> > [3] Nair V, Bartunov S, Gimeno F, et al. Solving mixed integer programs using neural networks[J]. arXiv preprint arXiv:2012.13349, 2020.
> >
> > [4] Junttila T, Kaski P. Conflict propagation and component recursion for canonical labeling[C]//International Conference on Theory and Practice of Algorithms in (Computer) Systems. Berlin, Heidelberg: Springer Berlin Heidelberg, 2011: 151-162.
> >
> > [5] McKay B D, Piperno A. Nauty and traces user’s guide (version 2.5)[J]. Computer Science Department, Australian National University, Canberra, Australia, 2013.

---

> > > ### Comment · Reviewer_zedB · 2024-11-28
> > >
> > > Thank you for the additions addressing limitations (though I am unsure if such updates are allowed after submission), and for supplementing the results with metrics on objective values and constraint violations.
> > >
> > > Clarifying my question -- the authors have already addressed it by discussing practical methods for detecting symmetry subgroups efficiently. I am increasing my rating by 2.

---

> > > > ### Author Response · Authors · 2024-11-30
> > > >
> > > > We would like to sincerely thank you for your thoughtful review and for increasing the rating. Your constructive comments have been incredibly valuable to us.
> > > >
> > > > We have submitted a revised manuscript that includes the discussion of limitations, the supplementary results with more metrics, as well as responses to other reviewers' suggestions. We hope these revisions meet your expectations.
> > > >
> > > > **If possible, we would be very grateful if you could let us know if there are any remaining concerns or questions, as we noticed that the rating remains slightly negative (<6). We truly appreciate your insights and would be more than happy to address any further points.**
> > > >
> > > > Once again, thank you so much for your time, consideration, and valuable suggestions！

---

### Meta-Review · Area_Chair_KUqL · 2024-12-18

**Metareview:**

Symmetry in integer linear programs (ILPs) poses challenges for graph neural networks, which have recently emerged as a promising approach for solving ILPs and yet struggle to distinguish symmetric variables. This work addresses this by proposing an orbit-based feature augmentation scheme that groups symmetric variables and assigns augmented features from a discrete uniform distribution. Numerical simulations showcase the advantages of the proposed framework both in training and predictive performance.

Despite some differing opinions on the merits and execution of the ideas, many reviewers agree that the paper presents novel ideas and makes valuable contributions to the field. While the individual components of the proposed framework may not be novel, the paper meaningfully advances ILPs by combining these concepts in an innovative way.

**Additional Comments On Reviewer Discussion:**

There were some concerns regarding the clarity of the presentation, its relation to prior works, and various aspects of the experiments that the authors appear to have addressed in their rebuttal.

---

### Decision · Program_Chairs · 2025-01-22

Accept (Poster)